# H-NGPCA: Hierarchical clustering of data streams with adaptive number of clusters and adaptive dimensionality

**Nico Migenda**[1]*, **Ralf Möller**[2], **Wolfram Schenck**[1]

**1** Center for Applied Data Science Gütersloh, Bielefeld University of Applied Sciences and Arts, Bielefeld, Germany, **2** Computer Engineering Group, Faculty of Technology, Bielefeld University, Bielefeld, Germany

* nico.migenda@hsbi.de

## Abstract

We present H-NGPCA, a hierarchical clustering algorithm for data streams that integrates an adaptive unit number growth and local dimensionality control. Unlike existing algorithm, H-NGPCA combines the characteristics of centroid-based, model-based and hierarchical clustering. H-NGPCA builds a hierarchical structure of local Principal Component Analysis (PCA) units, where each unit is a hyper-ellipsoid whose shape is updated by a neural network-based online PCA method. The repositioning of each unit is handled by Neural Gas, a centroid-based clustering algorithm. In the hierarchical tree structure, new units are created in a branch if suggested by a splitting criterion. In addition, each unit determines its own dimensionality based on the data represented by the unit. In extensive benchmarks, H-NGPCA not only surpasses all competing online algorithms with adaptive unit numbers but also achieves competitive performance with state-of-the-art offline methods, reaching an average NMI = 0.87 and CI = 0.26. This demonstrates that H-NGPCA achieves both online adaptability and offline-level accuracy.

## 1 Introduction

Cluster analysis is a family of methods from the field of exploratory data analysis. The goal is to sort different objects into groups in such a way that the similarity, which is determined by a proximity measure, between two objects is high if they belong to the same group and low if they belong to different groups. Clustering algorithms have been developed since the 1970s [1]. During the early development phase it was assumed that the data set is static and permanently available for processing. However, in many current applications, data streams are received and processed continuously [2].

A data stream is defined as an infinitely long, chronologically sorted sequence that must be analyzed directly as it is received under the constraints of limited memory and computational resources. Formally, a data stream $\mathcal{S}$ is a sequence of $N$ data points $\mathbf{x}_1, \mathbf{x}_2, \ldots, \mathbf{x}_N$, that is, $\mathcal{S} = \{\mathbf{x}_i\}_{i=1}^{N}$ which is possibly unbounded ($N \to \infty$). Each

**Data availability statement:** All relevant data to reproduce the results (mean, std, graphics) are uploaded with the revision as supporting information files (variety of .mat files). They are all further available from the clustering benchmark database (https://doi.org/10.1007/s10489-018-1238-7).

**Funding:** This research was funded by the German Federal Ministry of Education and Research (BMBF) in the project VIP4PAPS (to W.S.).

**Competing interests:** The authors have declared that no competing interests exist.

data point is described by an n-dimensional attribute vector $\mathbf{x_i} = [x_i^j]_{j=1}^n$. Data streams unfold sequentially over time, with samples arriving in an unpredictable manner. The unknown size of the data stream requires data reduction techniques as it is impractical to store the entire data set. The rapid arrival of samples requires real-time processing, emphasizing the need for immediate responses. The evolving nature of the content of data streams requires adaptability to account for changing characteristics [3]. The results derived from data stream processing are often approximations as only one data point is considered at a time. Efficient memory utilization is critical given the limitations of stream processing. Concept drift, where the underlying patterns can change, require adaptive algorithmic components [4]. Data streams often have unexpected characteristics such as uncertainty that highlights the need for robust and flexible processing methods. In summary, data streams present a complex and dynamic challenge for clustering algorithms which can only be approached by developing algorithms with real-time adaptability [5].

## 1.1 Problem statement

Clustering algorithms for data streams require the assignment of hyperparameters such as the number of clusters, thresholds for density and spacing, decay rates, window lengths, and many more. Such hyperparameters must be set according to the input data at hand and directly affect the quality of clustering. While setting such parameters is also difficult in traditional clustering, data streams undergo changes that may cause clusters to emerge, disappear, merge, or split. As such, setting a fixed value without prior knowledge would bias the clustering model. Specifically, determining the optimal number of clusters is critical. When cluster characteristics, such as data distribution, are known a priori, techniques such as cross-validation can determine the cluster count; however, in most cases, knowledge of the input data is not available prior to execution. Whenever two or more local regions (each represented by a cluster model) start drifting towards or apart from each other, the number of cluster models has to be adjusted accordingly.

In brief, clustering streams cannot be done by using traditional algorithms for batch processing of data sets. Intuitively, it is not possible to stop the stream to perform analysis, or to indefinitely postpone results in order to have the time to perform offline clustering. So, the naive solutions would be (i) to try and buffer the stream for later processing, which is not possible for endless streams of data, or (ii) to take random samples, which reduces the data size while attempting to retain representativeness but may fail to capture the correct data distribution. These naive solutions clearly do not make best use of the time available and of the information that the stream contains.

## 1.2 Related work

We provide a compact overview of state-of-the-art online clustering algorithms and highlight their properties with respect to data stream clustering; an in-depth review of data stream clustering algorithms relevant in this work is shown in section 6. Based on the conventional taxonomy for clustering algorithms [6], clustering algorithms

can be classified into the following categories: (i) Hierarchical; (ii) Grid-based; (iii) Density-based; (iv) Partitioning; and (v) Distribution-based.

(i) Hierarchical clustering organizes objects into a hierarchical tree structure. The hierarchical tree structure allows to create different clusterings by exploring the tree at different levels. The advantage of this category is the tree structure, as its possible to expand or prune each branch independently. However, the hierarchical structure is susceptible to noise and outliers [7,8]. (ii) In grid-based algorithms, the input data is divided into grid cells that group the objects of the data stream into bins. Regardless of how much data is presented, it is represented with a constant number of grid cells. The resolution of the grids is a trade-off between accuracy and computational cost; this is generally defined by user-supplied hyperparameters. The handling of the grid structure limits many grid algorithms of this type to low-dimensional data sets [9]. (iii) In density-based algorithms, clusters are generated as dense regions separated by less dense regions so that they can represent arbitrarily shaped clusters. In density-based algorithms, the definition of a density region is crucial, which is defined by hyperparameters [10] (iv) Partitioning algorithms create partitions so that similar objects are in the same partition and dissimilar objects are in different partitions. Partitions can be defined by centroids, representative points, or nodes. The online variant of the well-known K-means method from this class was compared to hierarchical methods and could achieve better results [11]. However, partitioning algorithms are mostly limited to hyperspherical clusters and the number of clusters is given as a fixed hyperparameter [12]. (v) Distribution-based clustering is based on the approach of learning a model of data distribution that best fits the input data. One of the most important advantages of distribution-based algorithms is their property of noise robustness. The quality of such methods strongly depends on the prior knowledge of the underlying data distribution, which is often unknown.

Some algorithms combine approaches from the different groups, with the goal of minimizing the weaknesses of each group. For example, D-Stream [13] and MR-Stream [14] are classified in the literature as both grid [15] and density methods [16]. A two-step hierarchical K-means that first partitions the data into spherical groups and then merges these groups using hierarchical methods is proposed in [17]. An other hierarchical extension of K-means [18] uses a pairwise overlap between components to determine the optimal number of clusters, while preserving the fast computation property of K-means. Also, the NGPCA algorithm [19] which is extended in this work naturally combines the strengths of Neural Gas (partitioning) and PCA (model-based) with the goal of achieving better clustering.

More recently, hybrid approaches were proposed that attempt to enhance classical clustering algorithms with deep learning [20]. Before these developments, several non-linear dimensionality reduction methods had already been introduced to overcome the limitations of linear techniques such as PCA. Examples include Kernel PCA, Autoencoders, and manifold learning techniques such as t-SNE and UMAP, which are able to capture complex nonlinear structures in high-dimensional data. Building on this, so-called deep clustering approaches [21] combine feature learning and clustering within a joint deep-learning framework. However, this comes with the typical deep learning drawbacks of significantly higher computational costs, lower interpretability, and sensitivity to hyperparameters.

## 1.3 Contributions

An example of the learning process of our hierarchical neural gas principal component analysis (H-NGPCA) algorithm is shown in Fig 1 with the corresponding unit tree in Fig 2 to highlight the adaptive splitting process. Our contributions compared with existing work are:

- Our algorithm combines the characteristics of three clustering categories: Centroid-based clustering (Neural Gas), model-based clustering (Principal Component Analysis) and a hierarchical structure.
- We provide an algorithm that adaptively decides when a unit needs to be split without the need for a termination criterion.
- We incorporate a local adaptive dimensionality control, so that each region is approximated with the correct dimensionality.
- All components are completely online without the need of historical data, pre-training, and batch or offline components.

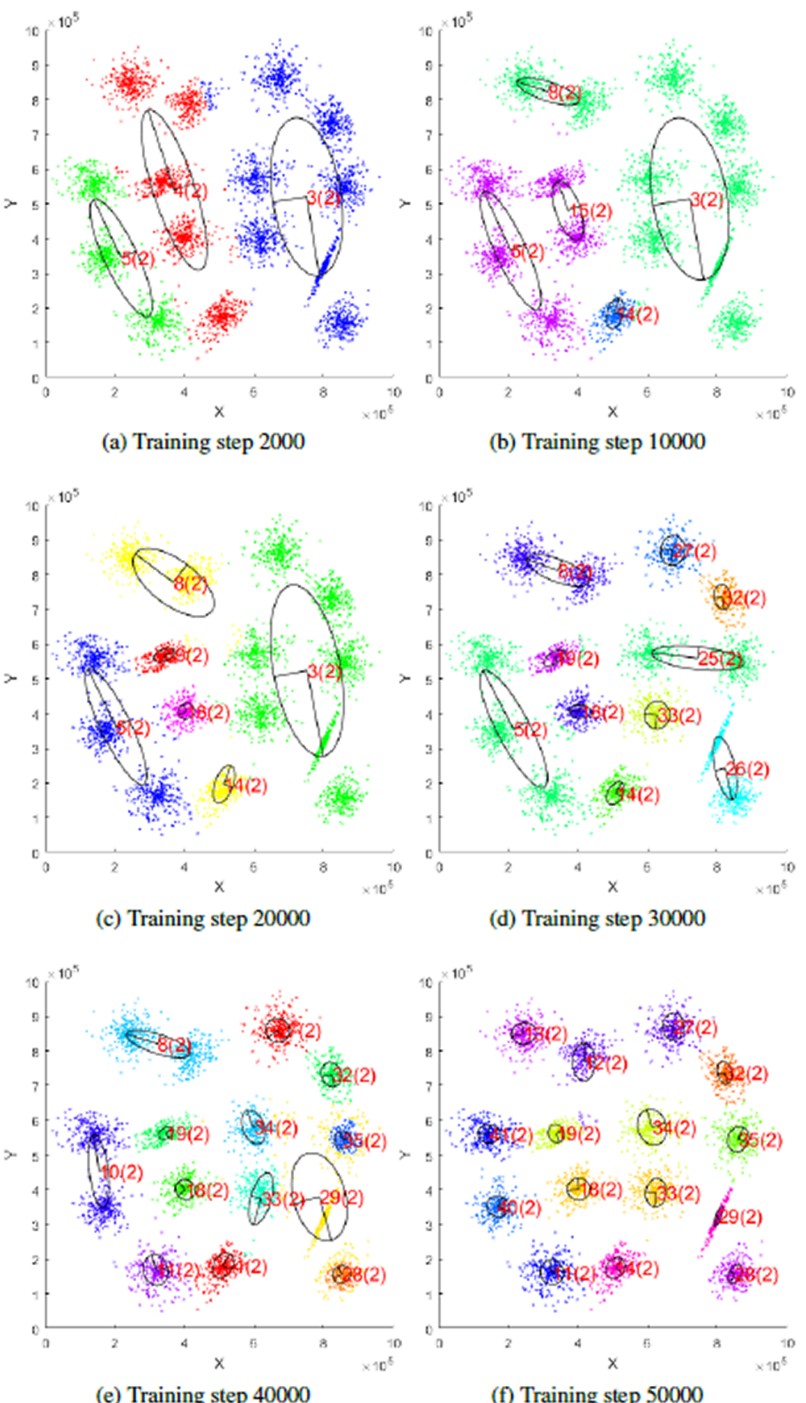

**Fig 1**. **Training of H-NGPCA on a two-dimensional data set with 15 clusters.** The pictures show the learning progress of H-NGPCA and the continuous splitting of the units. Each unit is indexed and the corresponding unit dimensionality is the value within the brackets. The corresponding unit tree is shown in Fig 2. After training step 50000, the number of units stays constant (tested up to step 75000.) We recommend to view this picture in color, as each color (evenly selected from the color spectrum) shows the assignment to the units.

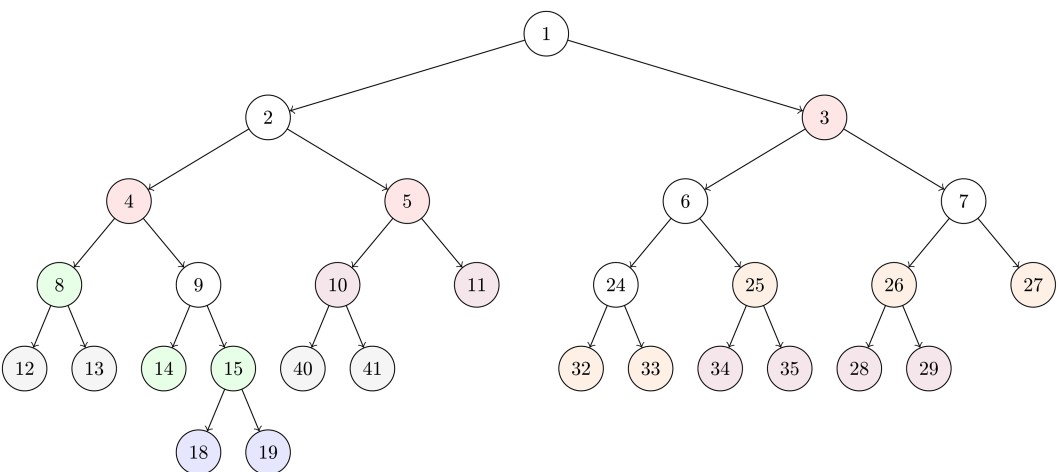

**Fig 2. H-NGPCA unit tree according to the clustering result of Fig 1.** The colors correspond to the units that are added at each of the six training snapshots (Fig 1(a)-(f)). After 2000 training steps the model consists of unit 3, 4 and 5 (red color). At training step 10000 (green color) the units 8, 14 and 15 emerged and replace unit 4. At training step 20000 (blue color) the units 18 and 19 emerged and replace unit 15. At training step 30000 (orange color) the units 25, 26, 27, 32 and 33 emerged from unit 3. At training step 40000 (purple color) the units 10 and 11 emerge from unit 5, 28 and 29 from unit 6, and the units 34 and 35 emerge from unit 25. In the final update after 50000 (gray color) training steps the units 12 and 13 emerge from unit 8 and the units 40 and 41 from unit 10. White units were created between two snapshots and split up again.

- We provide a fully reproducible benchmark and online repository, which can be used as a basis for future benchmarks in the field of data stream clustering with an adaptive number of units.

## 2 Local online PCA clustering

Classical offline Principal Component Analysis (PCA) preserves the maximal variance of a data set with a given set of linear descriptors. PCA performs an orthonormal transformation on a *n*-dimensional data set to obtain a smaller set of *m* linearly independent variables. As the focus of this work are online PCA algorithms, we refer to [22] for a detailed description of offline PCA.

A streaming or online setting for PCA is characterized by data points arriving sequentially over a period of time [23]. In each iteration *i*, one data point is presented and the model is parameters are updated. To maintain a good approximation, the set of parameters describing the subspace has to be updated continuously without access to the history of data. Different learning rules for online PCA were proposed [24], of which incremental and neural-network algorithms are the most popular. In the following we focus on neural-network-based PCA and refer to [25] for a review of incremental PCA.

In a mixture of local PCA, dimensionality reduction is combined with vector quantization (VQ). This is achieved by extending the simple codebook vectors by local PCA units, where each unit competes for the presented data points. Neural Gas Principal Component Analysis (NGPCA) is an online local PCA algorithm [19,22]. A NGPCA network consist of $M$ local units $\{\mathbf{c}_j, \mathbf{W}_j, \mathbf{\Lambda}_j, \check{\lambda}_j, \epsilon_j\}_{j=1}^M$ where the set of parameters of the *j*th unit is defined as $\theta_j$ and $\Theta = \{\theta_j\}_{j=1}^M$. The center points $\mathbf{c}_j$ are updated according to the Neural Gas (NG) scheme

$$\mathbf{c}_j \leftarrow \mathbf{c}_j + \epsilon_j \cdot h_\rho \left[ R_j(\mathbf{d}) \right] \cdot (\mathbf{x}_i - \mathbf{c}_j) \tag{1}$$

with $h_\rho \left[ R_j(\mathbf{d}) \right] = \exp\left( \frac{1 - R_j(\mathbf{d})}{\rho} \right)$ being an exponentially declining term that ensures that not only the winner is updated (soft-clustering). The ranking of each unit $R_j(\mathbf{d})$ is determined based on the distances $\mathbf{d} = [d_j]_{j=1}^M$, with rank 1 indicating the closest and rank $M$ the furthest Euclidean distance from a codebook vector to the presented data point $d_j = ||\mathbf{x} - \mathbf{c}_j||$.

Typically, the learning rate $\epsilon_j$ decreases over time. To stabilize online learning on dynamic distributions, the learning rate must be able to increase again later in the training process, so that the units can adapt to the new data. In the adaptive version [22] the adaptive learning rate $\epsilon_j$ is updated by

$$\epsilon_j(t) = \frac{1}{m} \sum_{i=1}^{m} |\bar{\Gamma}_{j,i}(t)| \tag{2}$$

with $\bar{\Gamma}_{j,i}(t)$ being an adaptive term that depends on the unit fit to their representing data points and is approximated in a low-pass filter [22]. This adaptive term is calculated for each of the $m$ dimensions and then averaged. The neighborhood range $\rho$ as typically used in Neural Gas is updated globally for all units. It is determined from the average learning rate $\epsilon_j(t)$ of all $M$ units:

$$\rho = \frac{1}{M} \sum_{j=1}^{M} \epsilon_j(t). \tag{3}$$

The eigenvalues $\Lambda_j$, eigenvectors $\mathbf{W}_j$ and the mean residual variance $\check{\lambda}_j$ in the $n$–$m$ minor eigendirections are recursively obtained by a coupled neural PCA learning rule [26].

From a geometric perspective, each local PCA unit is a hyper-ellipsoid located in the input data space (Fig 3). For each presented data point it has to be determined to which local model(s) the data point belongs. A ranking $R_j(\mathbf{d})$ based on a Mahalanobis distance measure is performed, where the Mahalanobis distance is one possible choice [19]. The Mahalanobis distance is defined as (unit index $j$ omitted)

$$d_M(\xi) = \xi^T \mathbf{W} \Lambda^{-1} \mathbf{W}^T \xi + \frac{1}{\check{\lambda}} \xi^T \left( \mathbf{I} - \mathbf{W} \mathbf{W}^T \right) \xi \tag{4}$$

with $\xi = \mathbf{x} - \mathbf{c}$; the second additive term only occurs for $m < n$ as $\check{\lambda} = 0$ for $m = n$. As only units with a good ranking are updated, it may happen that some units are not winning any data points and are then considered "dead".

In [27] an alternative distance measure $d_H(\xi)$ (the index H identifies the author's name) is proposed to prevent dead units. The eigenvalue and eigenvector estimates are updated by the same PCA procedure, but the distance measure used in the ranking is independent of the volume $V$ of the hyper-ellipsoid corresponding to a unit:

$$d_H(\xi) = \left( \xi^T \mathbf{W} \Lambda^{-1} \mathbf{W}^T \xi + \frac{1}{\check{\lambda}} \xi^T \left( \mathbf{I} - \mathbf{W} \mathbf{W}^T \right) \xi \right) V^{\frac{2}{n}}, \tag{5}$$

with $V = \sqrt{|\Lambda| \check{\lambda}^{n-m}}$ and $\xi = \mathbf{x} - \mathbf{c}$. This potential function is essentially treating each unit as a hyper-ellipsoid of the same volume in the competition, regardless of its actual volume.

## 3 Hierarchical online local PCA clustering

From the viewpoint of hierarchical local PCA clustering, such as NGPCA (section 2), each unit has the same horizontal position in a single layer (heterarchy) and same properties, such that each unit theoretically plays an equal role during the ranking process. In hierarchical NGPCA clustering (H-NGPCA) [28], units are placed on multiple horizontal layers with units closer to the root having a larger share of data points. A hierarchical structure and the data flow of H-NGPCA are shown in Fig 4(a). The model consists of a binary tree of PCA units, with all units belonging to the set $\mathcal{U}$. Within the tree, we distinguish between two different kind of units. Firstly, there are the "developed" units $\mathcal{U}_d \subset \mathcal{U}$, which have two so-called child units directly connected to them in the binary tree structure. The child units are one hierarchical level lower in

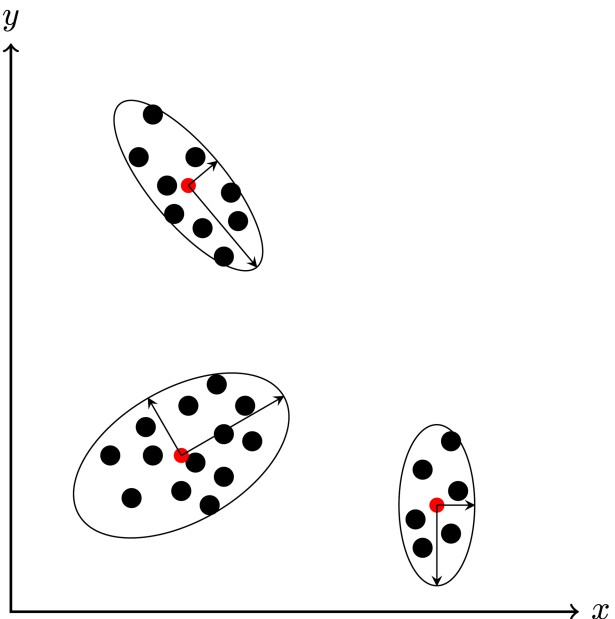

**Fig 3. Simple clustering problem with three clusters.** Each cluster is represented by a PCA unit with an axis length of $\sqrt{\lambda_i}$ and a specified center (red dot). Black dots represent data points.

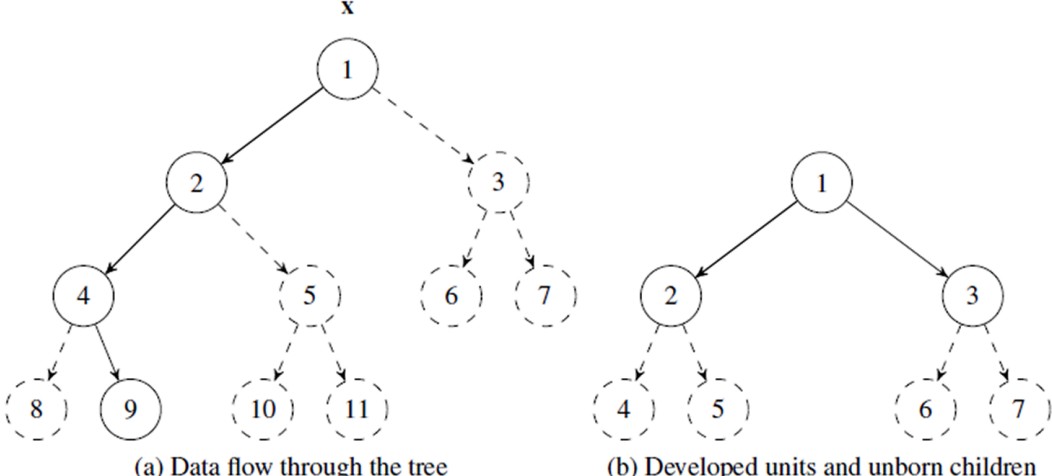

(a) Data flow through the tree  (b) Developed units and unborn children

**Fig 4. Hierarchical structure of H-NGPCA.** (a) On each data point presentation, the presented data point **x** is passed trough the winning branch until an unborn children is reached. The flow of data point **x** is indicated by the solid line path. Respectively, the winner units have a solid shape and all loser units a dashed shape. (b) Units with a solid shape are developed units (set $\mathcal{U}_d = \{1, 2, 3\}$) which have children. The dashed units on the lowest level are unborn children (set $\mathcal{U}_u = \{4, 5, 6, 7\}$). They are trained normally and compete against their respective parent unit to become a developed unit with its own unborn children.

the branch and the child units can in turn also have children, which gradually builds up the tree. This structure continues to the last hierarchical level of each branch until it reaches the units that no longer have children of their own. These units are referred to as unborn children or unborn units $\mathcal{U}_u \subset \mathcal{U}$. Unborn children have the same properties as developed units and are fully involved in the learning process; the only difference is that they have no children themselves. This is shown

in Fig 4(b), with the solid-drawn units representing $\mathcal{U}_d$ and dashed-drawn units $\mathcal{U}_u$. The unborn units $\mathcal{U}_u$ are trained normally and their goal is to outperform their parent unit (from which they were generated) so that they themselves become developed units, which in turn each receive two unborn children of their own. For the later derivation of this competition of the unborn children against their parents, we introduce the set $\mathcal{U}_b \subset \mathcal{U}_d$, which contains the outermost developed unit of each branch.

Two example cases are visualized in Fig 5. In Fig 5(a), a root unit (solid line) is shown that lies between two clusters. The two unborn child units (dashed line), on the other hand, represent the two clusters well, which indicates a split. Another scenario is shown in Fig 5(b), where the model consists of the three clusters represented by the three outermost developed units. Then, the data points of the bottom left cluster (orange squares) start to drift apart, and the two children (dashed line) follow. If the newly created clusters drift apart a little further, a split is suggested. The two other higher-level units (solid line) also have two unborn child units each, which are not drawn for reasons of clarity.

## 3.1 Algorithm overview

To provide a intuitive description of H-NGPCA, we will provide a step-by-step explanation of the training procedure on each data point presentation. A full visual summary of all steps is shown in Fig 6 and the pseudo code is included in the appendix B.

The H-NGPCA model is initialized with one root unit which can be regarded as a global PCA and two corresponding unborn children. On each data point presentation, the set of winner units ($\mathcal{W} = \{\}$) and looser ($\mathcal{L} = \mathcal{U}$) units is reset. In a top-down approach (line 7-17 in algorithm 2) the data point is passed through the tree, starting with the two children of the root unit. A pairwise ranking is performed between the two child units. For the pairwise ranking, first, the potential of both units ($[d_1(\xi), d_2(\xi)]$) to the presented data point are calculated based on (5). The winner unit is then determined by comparing which one has the lower potential

$$c_w \leftarrow \arg\min(d_1(\xi), d_2(\xi)) \tag{6}$$

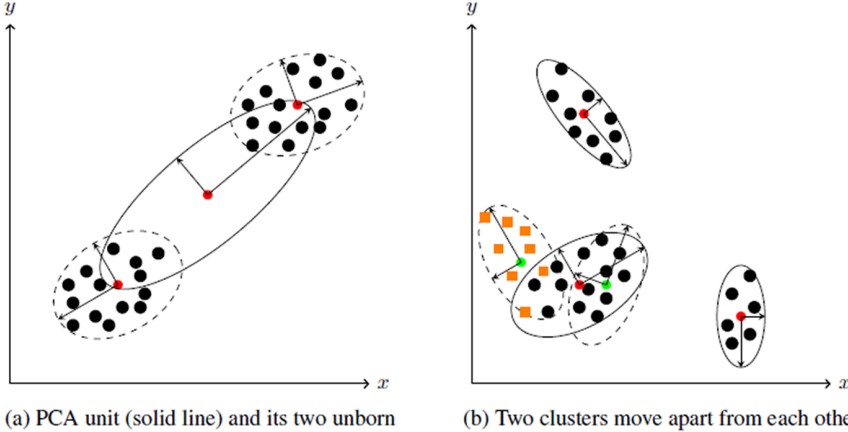

(a) PCA unit (solid line) and its two unborn children (dashed line)

(b) Two clusters move apart from each other sugesting a split operation

**Fig 5. (a): a PCA unit (solid line) lies between two distributions, poorly approximating them.** The two unborn units (dashed lines) are much better approximations of the distributions, suggesting a split operation. (b): The original data points (dots) form three clusters. With the presentation of new data points (rectangles) over time, a cluster is splitting into two clusters. The unborn units (dashed line) yield a better approximation of the new distribution.

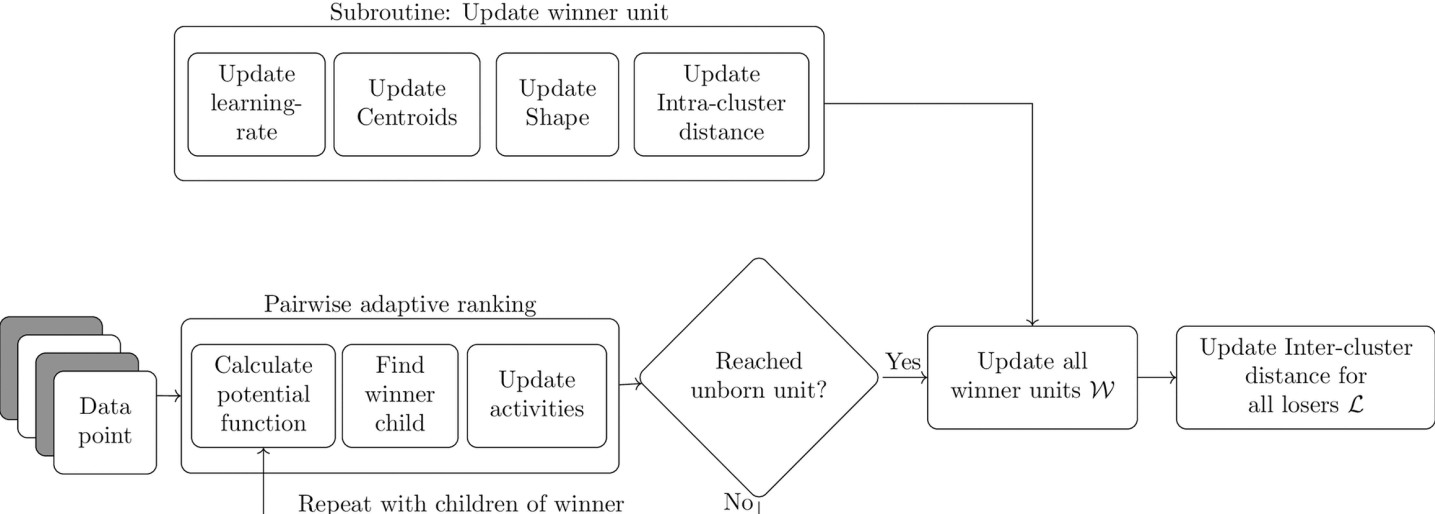

**Fig 6**. **Sketch of the proposed H-NGPCA method.** The method is divided into two phases; this figure covers the update procedure of the model parameters; Fig 7 covers the adjustment of the model size. Starting from the root unit, the current data point is used to calculate the potential function of both child units. This competition leads to a winner unit. While both units get their activities updated, only the winner unit has its model parameter updated. This includes the learning rate, centroids, shape and intra-cluster distance. If children exist, the processes is repeated. Then, the inter-cluster distance for all loser units $\mathcal{L}$ is updated for the given data point; the winner units already got their intra-cluster distance updated.

and the winner is added to the set of units that have their parameters updated $\mathcal{W} \leftarrow \mathcal{W} \cup c_w$. This procedure is recursively repeated as long as the winner unit $c_w$ has its own children. In this way, the data point is passed through one branch of the tree until it reaches an unborn unit. This winner path is indicated with the solid line in Fig 4(a).

Once the winner branch is identified, all units in $\mathcal{W}$ are updated, in a winner takes all setting (hard-clustering), which means that the loser units are unchanged. This second part of the algorithm corresponds to line 18-25 in algorithm 2. All winner units have their learning rate (2), centers, eigenvalues and eigenvectors [26], assignment value, as well as their activity and intra-cluster distance updated. The intra-cluster distance is a key component for the unit splitting algorithm which is discussed in detail in sec. 3.2. The asignment value is always determined between two siblings. It describes the proportion to which the data points are divided between the two units. This is necessary in order to correctly incorporate the weighting when comparing two child units with their parent unit. The activity describes the weighting of the outermost developed units. This is necessary to weight the quality of the model $\mathcal{U}_b$. The update steps for the above parameters are described below.

The unit centers are updated according to

$$\mathbf{c}_j \leftarrow \mathbf{c}_j + \epsilon_j \left( \mathbf{x} - \mathbf{c}_j \right), \tag{7}$$

which is similar to the update in NGPCA (1), except that there is no ranking, as the winner takes all. The assignment value is a weighting factor between two sibling units $j \in \{j_1, j_2\}$, that is updated for all units along the winner branch

$$a_j(\mathbf{x}) = \begin{cases} 1, & \text{if winner unit for } \mathbf{x} \text{ is } j \\ 0, & \text{otherwise.} \end{cases} \tag{8}$$

To approximate the overall assignment value over all data points, a low-pass filter is used

$$\bar{a}'_j \leftarrow (1 - \mu)\bar{a}_j + \mu a_j(\mathbf{x}) \quad \forall \quad j \in \{j_1, j_2\} \tag{9}$$

with $\mu$ being a low-pass parameter. The assignment values of both units are then normalized

$$\bar{a}_{j_1} \leftarrow \frac{\bar{a}'_{j_1}}{\bar{a}'_{j_1} + \bar{a}'_{j_2}}, \quad \bar{a}_{j_2} \leftarrow \frac{\bar{a}'_{j_2}}{\bar{a}'_{j_1} + \bar{a}'_{j_2}} \tag{10}$$

so that $\sum_{j \in \{j_1, j_2\}} \bar{a}_j = 1$. The assignment value is important for the splitting procedure, as it defines to what proportion the two units replace their respective parental unit (next higher hierarchical level and directly connected). Once the winner units $\mathcal{W}$ are updated, in all other units (Fig 4: dashed units) the inter-cluster distance is updated. For high-dimensional data an additional component exists for the winner units, namely an update of the unit specific dimensionality $m_j$. Once the model parameters are updated, it is checked whether the tree should be extended or not (sec. 3.3). The detailed algorithms for both the dimensionality and unit-number adjustment are discussed in separate algorithms below. This procedure is repeated on each data point presentation to continuously update the model parameters.

### 3.2 Quality measure for model selection

Determining the optimal number of clusters in a data set is a fundamental issue in partitioning clustering. Some algorithms such as K-means clustering, require the user to specify the number of clusters k to be generated. Unfortunately, the optimal number of clusters is subjective and depends on the method used for measuring similarities and the parameters used for partitioning. So this task should not be left to the user.

In hierarchical clustering algorithms, a quality measure is derived based on a suitable distance and a linkage criterion that specifies the dissimilarity of the data points belonging to a unit. H-NGPCA is a deterministic and geometrical clustering approach that is based on a version of the Mahalanobis distance metric such as (4) or (5) to determine the similarity between data points and group them into clusters. Therefore, we will use the geometric properties to define a quality measure that determines whether a pair of unborn child units outperforms its respective parent unit, resulting in a split.

The model is represented by the outermost developed unit of each branch $\mathcal{U}_b$. Each of these outermost developed units have two unborn child units, all unborn units are gathered in the set $\mathcal{U}_u$, that compete to replace the respective parent unit. On each data point presentation, the distance of all these units to the presented data point is calculated (5). In order to approximate a continuous quality measure and since it is impossible to calculate the sum over all data points $N$, the distances are approximated by a low-pass filter. As the classical Mahalanobis distance (4) is biased towards large units [22,27], we consider the volume-normalized Mahalanobis distance $d_H(\xi)$ (5)

$$\bar{d}_{H,j} \leftarrow (1 - \mu)\bar{d}_{H,j} + \mu d_{H,j} \quad \forall \quad j \in \mathcal{U}_b \cup \mathcal{U}_u \tag{11}$$

with a low-pass parameter $\mu$. The distances $\bar{d}_{H,j}$ of all unborn child units $\mathcal{U}_u$ and their respective parent units $\mathcal{U}_b$ are necessary for the split decision. Further, an activity between the units $\mathcal{U}_b$ is necessary, so that the distances of the individual units can be weighted into a quality measure; equal weighting would not work with unbalanced data. For this purpose, an activity $\bar{\pi}$ is introduced, which defines the weight between all units $\mathcal{U}_b$ that currently represent the model. In analogy to the assignment value $\bar{a}$ (see (8)-(10)), this leads to

$$\pi_j(\mathbf{x}) = \begin{cases} 1, & \text{parent of } c_w \\ 0, & \text{otherwise.} \end{cases} \tag{12}$$

with only one unit of $\mathcal{U}_b$ (parent of $c_w$) obtaining a $\pi_j = 1$ and all other units $\pi_j = 0$. Then, for each unit the continuous activity is approximated in a low-pass filter

$$\bar{\pi}_j' \leftarrow (1 - \mu)\bar{\pi}_j + \mu\pi_j(\mathbf{x}) \quad \forall\, j \in \mathcal{U}_b \tag{13}$$

and normalized over all outermost developed units

$$\bar{\pi}_j = \frac{\bar{\pi}_j'}{\sum_{i \in \mathcal{U}_b} \pi_i}, \quad \sum_{j \in \mathcal{U}_b} \bar{\pi}_j = 1. \tag{14}$$

It is now possible to create a quality measure using the low-pass filtered distances $\bar{d}_{H,j}$ and activities $\bar{\pi}_j$.

In hierarchical clustering, a linkage criterion is commonly used to specify the similarity or respectively the dissimilarity between data points and clusters. In our method, each unit updates an intra-cluster and an inter-cluster distance. The similarity of data points that belong to a unit is tracked by the intra-cluster distance. All winner units $\mathcal{W}$ of $\mathcal{U}_b \cup \mathcal{U}_u$ have their intra-cluster distance updated in a low-pass filter

$$\bar{d}_{\mathrm{intra},j} \leftarrow (1 - \mu)\bar{d}_{\mathrm{intra},j} + \mu d_{H,j}(\boldsymbol{\xi}) \quad \forall\, j \in \mathcal{W} \cap (\mathcal{U}_b \cup \mathcal{U}_u) \tag{15}$$

with the low-pass parameter $\mu$, the set of winner units $\mathcal{W}$, the set of outermost developed units $\mathcal{U}_b$ and the set of unborn child units $\mathcal{U}_u$. The intra-cluster distance represents the average distance to all data points won by a unit. It remains unchanged for all loser units $\mathcal{L}$.

The inter-cluster distance is a natural counterpart that represents the average distance to all data points which do not belong to the unit. It is updated in a similar style as the intra-cluster distance but for all loser units $\mathcal{L}$ according to

$$\bar{d}_{\mathrm{inter},j} \leftarrow (1 - \mu)\bar{d}_{\mathrm{inter},j} + \mu d_{H,j}(\boldsymbol{\xi}) \quad \forall\, j \in \mathcal{L} \cap (\mathcal{U}_b \cup \mathcal{U}_u) \tag{16}$$

with $\mu$ being the low-pass parameter, the set of winner units $\mathcal{W}$, the set of outermost developed units $\mathcal{U}_b$ and the set of unborn child units $\mathcal{U}_u$. It remains unchanged for all winner units $\mathcal{W}$.

Based on the continuously updated intra-cluster and inter-cluster distance and the units activity, we can define an adaptive parameterless quality measure

$$Q(\theta \,|\, \mathcal{S}) = \sum_{j=1}^{\mathcal{U}_b} \bar{\pi}_j \frac{\bar{d}_{\mathrm{intra},j}}{\bar{d}_{\mathrm{inter},j}} \tag{17}$$

based on the sum of the activity-weighted ratio between intra-cluster (15) and inter-cluster (16) distance. Ideally the intra-cluster distance $\bar{d}_{\mathrm{intra}}$ is small when the represented data points are close to the respective unit. The inter-cluster measure $\bar{d}_{\mathrm{inter}}$ on the other hand should be large, as data points that do not belong to a unit are ideally further away from it. The activity weighting is necessary for unbalanced data sets, such that units that only represent a small share of the presented data are not distorting the quality measure.

### 3.3 Splitting algorithm

The next step is to use the updated model parameters (Fig 6) to check whether the model structure should be extended or not. The splitting algorithm is shown in Fig 7. First, the quality of the current model is calculated with (17). All units of $\mathcal{U}_b$ are taken into account, i.e. all units that have unborn children. Then, successively each unit of $\mathcal{U}_b$ is substituted by its two

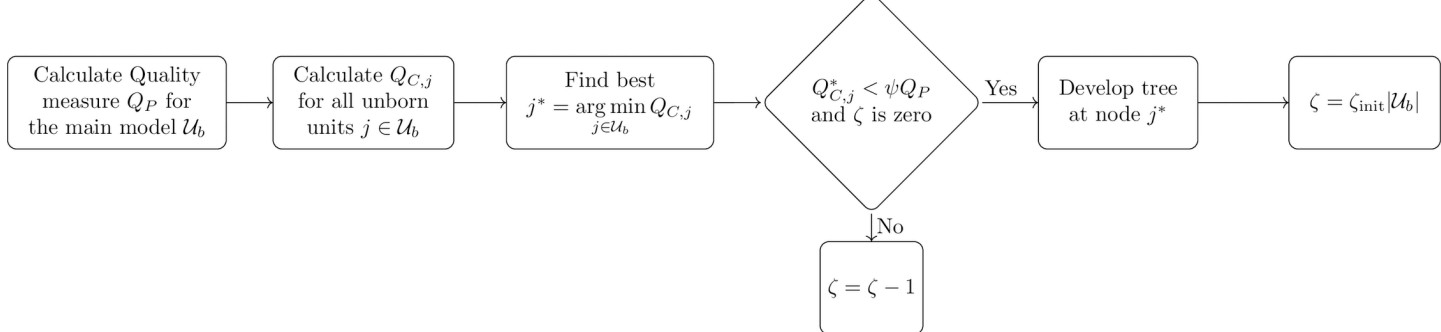

**Fig 7**. **With the updated model parameters (Fig 6) the quality measure for the current model $Q_P$ can be calculated.** In addition, the measure is calculated by substituting one developed unit by its unborn children $Q_{C,j}$. If the unborn children outperform their respective parent unit, a split is performed. The winning unborn units are upgraded to a developed unit and each obtains two new unborn units.

unborn child units; only one unit is replaced per comparison. Now we have the quality of the current best model $\mathcal{U}_b$ and the set of qualities where each time a single unit is substituted by its two children. To give a small example: For a model with ten outermost developed units, there are now eleven quality measures. One is the original measure $Q_P$ obtained from $\mathcal{U}_b$ and ten versions $Q_{C,j}$ in which always one outermost developed unit is replaced by its two unborn children. The intra-cluster and inter-cluster distance of parent unit $j$ is replaced by

$$\bar{d}_{\text{intra},j} \leftarrow \bar{a}_{j_1} \bar{d}_{\text{intra},j_1} + \bar{a}_{j_2} \bar{d}_{\text{intra},j_2} \tag{18}$$

$$\bar{d}_{\text{inter},j} \leftarrow \bar{a}_{j_1} \bar{d}_{\text{inter},j_1} + \bar{a}_{j_2} \bar{d}_{\text{inter},j_2} \tag{19}$$

with $\sum_{j \in \{j_1, j_2\}} \bar{a}_j = 1$ to weight the unborn units measures. From all eleven quality measures the minimum is searched $Q_{C,j}^*$. Then, it is checked if $Q_{C,j}^*$ is smaller than $Q_P$. To prevent false splits due to statistical outliers, we introduce a hysteresis parameter $\psi \in [0, 1]$ to slightly penalize the child models, so that $Q_{C,j}^* < \psi Q_P$. Throughout the benchmark, a value close to 1 is chosen for $\psi$. If the original measure $Q_P$ remains the best, no splitting takes place. Otherwise the tree is developed in the best branch, by replacing one unit by its children. After a split is performed, the model can not split for $\zeta$ training steps, to allow the newly added units to reposition. To ensure enough training time even with a large model size, the split prevention parameter grows with the model size $\zeta = \zeta_{\text{init}} |\mathcal{U}_b|$. As the splitting affects the entire model, this parameter is global.

If a split decision is made, the tree is developed further at the given location. This means that the two previously unborn children now become independent units and are added to $\mathcal{U}_d$ and $\mathcal{U}_b$. The respective parent unit of the now developed units is removed from $\mathcal{U}_b$, as it is not the lowest developed unit on this branch anymore. The two new developed units each obtain two new unborn children; these are respectively added to the set $\mathcal{U}_u$. Due to the hierarchical structure, the newly initialized children are always located in the subspace of the parent unit. The dimensionality is set to $m = 2$. Starting with a low dimensionality has a computational advantage. If a unit needs a higher dimensionality through the training process, this is achieved with our adaptive dimensionality control algorithm. The eigenvalues are set to half of the parent units first two eigenvalues, the mean residual variance accordingly, the centers and the first two principal eigenvectors are set to the values of the parent units. This leads to two identical children after initialization, so that the first competition between them is random, and the intra- and inter-cluster distances are set to 1. The learning rate is set to $\epsilon = 0.99$ so that the unit starts actively. If it were to inherit the learning rate of the parent unit, it would possibly start inactive and would first have to be woken up using an adaptive learning rate. The low-passes to calculate the learning rate are inherited from

their parent unit, but with a small offset of 10%. If the low-passes were adopted exactly, the calculation of the learning rate would directly result in the learning rate of the parent unit.

## 4 Local adaptive dimensionality adjustment

Each unit covers a different part of the data distribution, while each cluster possibly has a different dimensionality which furthermore constantly changes due to the continuous presentation of new data points. This requires each PCA unit to adaptively adjust its own dimensionality whenever the dimensionality of the represented cluster changes (algorithm 1). The need for a unit-specific dimensionality is further motivated by the hierarchical structure. This effect can be seen in Fig 8, as a parent unit lies between two clusters while the two corresponding child units lie correctly on one cluster each. The data set has 32 dimensions, and for the two children to represent 50% of the cluster variance, 7 and 10 dimensions are required respectively. One might expect the parent unit to require a similar dimensionality, but this is not the case. The parent unit does not represent the variance of the two clusters, but mostly the distance between them, which requires only two dimensions to represent 99% of the variance. The shape even suggests that if the distance is large enough, one dimension would be sufficient. This implies the need to adjust the dimensionality at each unit individually for each data point presentation, so that the best possible fit is achieved.

**Algorithm 1 Local online PCA dimensionality adjustment procedure.**

**Input:** current dimensionality $m_j$, current eigenvectors $\mathbf{W}_j$, current eigenvalues $\mathbf{\Lambda}_j$, current $\sigma_j^2$, current $\gamma_j$ with an initial $\gamma_j = \gamma_0$
**Output:** updated dimensionality $m_j$, updated eigenvectors $\mathbf{W}_j$, updated eigenvalues $\mathbf{\Lambda}_j$, updated $\sigma_j^2$, updated $\gamma_j$

```
 1: if γⱼ == 0 then
 2:    Ṽ ← Log Transformation (diag⁻¹(Λ))                          ▷ [25] Eq (15)
 3:    a, b ← Linear Regression (Ṽ)                                ▷ [25] Eq (16)
 4:    Ũ ← Log Eigenvalue Estimation (a, b)                        ▷ [25] Eq (17)
 5:    U ← Linear Transformation (Ũ)                               ▷ [25] Eq (18)
 6:    mⱼ ← Stopping Rule (U)                                      ▷ [25] Eq (20-23)
 7:    γⱼ = γ₀
 8:    if mⱼ increased then
 9:       Extend Λⱼ by the number of added dimensions from U
10:       Decrease residual variance σⱼ² by the added eigenvalues
11:       for all newly added dimensions do
12:          Wⱼ ← Modified Gram-Schmidt(Wⱼ)                        ▷ (20), (21)
13:       end for
14:    else if mⱼ decreased then
15:       Remove dimensions from Λⱼ
16:       Remove dimensions from Wⱼ
17:       Increase residual variance σⱼ² by the removed eigenvalues
18:    end if
19: else
20:    γⱼ ← γⱼ − 1
21: end if
```

For this purpose, we adapt the method from [25], in which an adaptive dimension adjustment for a PCA unit was presented. In the following, we describe the basic functionality of the dimensionality adjustment algorithm, but focus on significant extensions in this work and refer to the basic version of the algorithm for a detailed explanation of the dimensionality adjustment algorithm. The extended local adaptive dimensionality adjustment algorithm is shown in algorithm 1.

The dimensionality adjustment algorithm [25] exploits several natural features of neural network-based PCA and properties of the data distribution: (i) the eigenvalues are naturally sorted in a descending order, (ii) the components

 

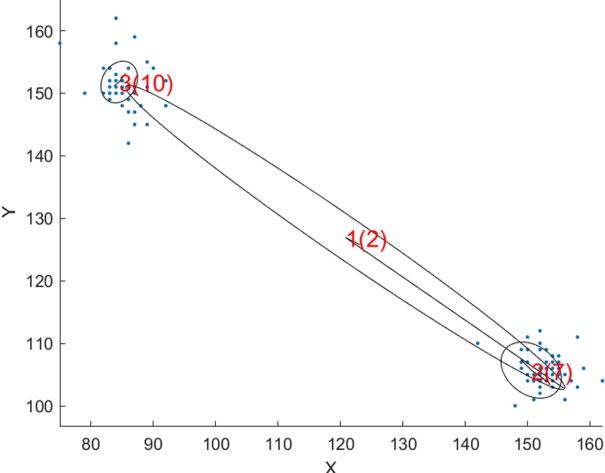

**Fig 8**. **Two clusters of a 32-dimensional data set represented by a parent unit and the two corresponding children.** The first number next to each unit center represents the unit index and the number in the bracket the corresponding unit dimensionality.

are trained in a hierarchical order, ensuring that the most relevant component is trained first, and (iii) the variance is not evenly distributed over all principal components. It is more likely that only some represent a majority of the data variance and the variance of most principal components is minor.

Each unit is initialized with a dimensionality of $m = 2$ so that only the two most relevant principal components are trained, which greatly reduces the computational effort. A least-square regression on a logarithmic scale is used to approximate the remaining $n-m$ eigenvalues. For this purpose, the $m$ eigenvalues are transformed into the logarithmic scale and the least squares parameters of a fitted line are estimated. The original $m$ eigenvalues are then supplemented with $n-m$ estimated log eigenvalues along the fitted line and transformed back into the non-logarithmic range. This allows to add or remove multiple dimensions at once if necessary and is the main reason why we chose this method. In the following we propose three adjustments to the algorithm:

(i) In the original algorithm [25], the corresponding eigenvectors to the approximated eigenvalues are selected at random, which is not ideal as the randomly generated eigenvectors can point into the space already covered by the first $m$ principal components. Instead, we use the modified Gram-Schmidt algorithm to calculate an orthonormal vector for each added dimension which is then used to extend the already existing eigenvector matrix. As the already existing eigenvectors in **W** are orthonormal to each other, it is only necessary to orthonormalize the newly added eigenvectors to the already existing ones. Therefore, for each added dimension the Gram-Schmidt procedure is performed sequentially. First, a random new eigenvector $\mathbf{w}_{m+1}$ is initialized. Then, we modify $\mathbf{w}_{m+1}$ according to

$$\mathbf{w}_{m+1} \leftarrow \mathbf{w}_{m+1} - \sum_{i=1}^{m} \mathbf{w}_i(\mathbf{w}_i^T \mathbf{w}_{m+1}) \tag{20}$$

normalize the new vector to unit length

$$\mathbf{w}_{m+1} \leftarrow \frac{\mathbf{w}_{m+1}}{\|\mathbf{w}_{m+1}\|} \tag{21}$$

and extend the eigenvectors matrix **W** by the newly approximated dimension. This process is repeated until the added eigenvector is orthogonal with sufficient accuracy.

(ii) The neural network-based PCA method approximates at each data point presentation the variance $\check{\lambda}$ in the $n$–$m$ minor eigendirections. When the dimensionality is adjusted, the approximation of $\sigma^2$ has to be adjusted. When the dimensionality is increased, the newly added variance (represented by the added eigenvalues) has to be subtracted from $\sigma^2$, and when the dimensionality is decreased, the variance needs to be added, respectively. This is not done in the original version, which results in a distortion of the potential function (e.g. (4) and (5)).

(iii) In the original algorithm, an initial delay of $\Gamma$ training steps is defined to allow the PCA to adapt to the data after a change in dimensionality. As we are now working with a mixture of local PCA, we extend this concept. First, whenever a unit adjusts its own dimensionality, we introduce a unit-specific delay $\gamma_j$ that prevents further dimensionality changes for that unit. Therefore, whenever new units are generated they also have the initial delay $\gamma_0$ to prevent uncontrolled changes during the unit adjustment.

## 5 Experiments

**Data sets:** The experimental study conducted in this paper is based on the data sets provided in the clustering benchmark database [29], appendix C.

**Baselines:** We benchmark our algorithm against state-of-the-art clustering algorithms, which we all re-implemented using popular python libraries, apendix D.

**Metrics:** To evaluate the clustering performance we use the commonly used Cluster Index (CI) and Normalized Mutual Information (NMI) measures, appendix C.

### 5.1 Visual results

H-NGPCA was tested on all data sets (Table 4). To get a first impression of the performance, selected clustering results are visualized and discussed. In order to limit the scope to the essential results, the majority of the visual results are discussed in the apendix E. The figures serve to visually validate the numerical results of the following chapters. For referencing between the visualization and the text, each unit is numbered with the dimensionality indicated in brackets. In addition, the data points are colored based on the clustering results, so the images in this chapter should be viewed in color. The final cluster results are shown with the associated time course of quality $Q(t)$, learning rate $\epsilon(t)$, number of units, and dimensionality $m(t)$ during the learning process. For the high-dimensional data sets, the PCA units are shown by their two most relevant principal components.

In Fig 9, the H-NGPCA algorithm is trained on the s1 data set. This initial data set is characterized by 15 two-dimensional Gaussians without overlap. Each cluster is covered by exactly one unit and the assignment of the data points works without errors. The learning process on that data set is shown in Fig 1.

While the previous visualizations concerned 2d data sets, now the high-dimensional data set h1024 is considered. This example is particularly relevant as it can be used to analyze the influence of the adaptive dimensionality control. The question arises as to whether drastically changing the dimensionality of the individual units leads to undesirable splits. Fig 10 shows the 2d projections of the units on the h1024 data set. The axes of the units are the 2 eigenvalues with the highest variance. The numbers in brackets after the unit index indicate the dimensionality of the respective unit. It can be seen that each unit has its own dimensionality based on the data associated with it. In addition, each unit is located on exactly one cluster and the data points are correctly assigned to the units. Fig 11.a shows the dimensionality of the units over time. We decided not to take averaged curves over multiple runs, as it cannot be guaranteed that each unit represents the same cluster in every simulation. Newly added units tend to overshoot slightly in their dimensionality before they converge towards the correct dimensionality. This also corresponds to the findings from [25]. It can be seen that the dimensionality converges very well against the actual value and that fluctuations in dimensionality have no influence on the quality measure and thus the splitting process. The ground truth dimensionality of all clusters which was determined

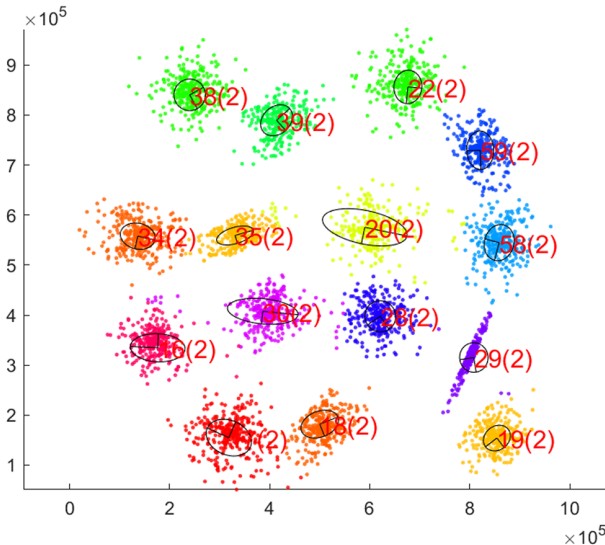

**Fig 9. Final clustering on the s1 data set.** All PCA units are represented with an axis length of $\sqrt{\lambda_i}$.

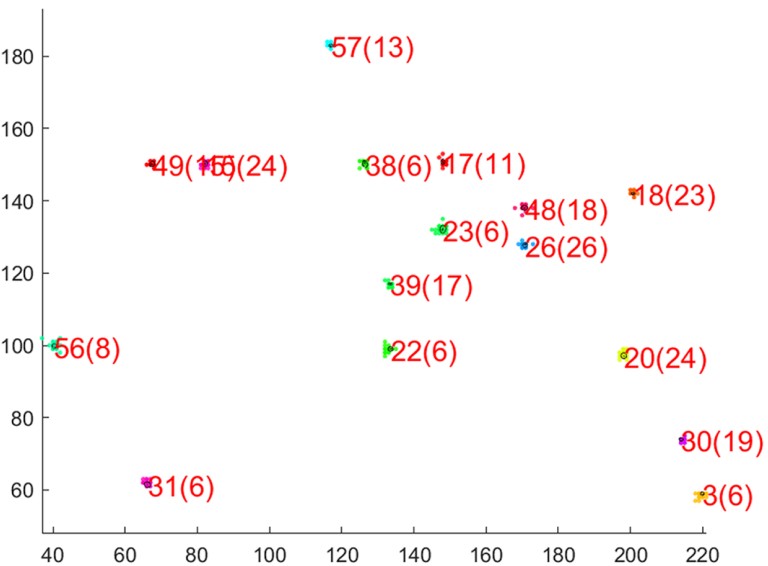

**Fig 10. Final clustering on the h1024 data set.** All PCA units are represented with an axis length of $\sqrt{\lambda_i}$.

offline is $m = 15.5$ and the average PCA dimensionality at the end of the training procedure is $m = 16.75$. Fig 11.b shows the continuous growth of outermost units until the real number of clusters is reached.

## 5.2 Quantitative and statistical comparison with competing algorithms

The literature search revealed that none of the online clustering algorithms with an adaptive number of units under consideration provided a functioning implementation. We therefore extended our benchmark to offline algorithms with an adaptive number of clusters and established popular offline clustering algorithms with a fixed number of units. The NMI and CI

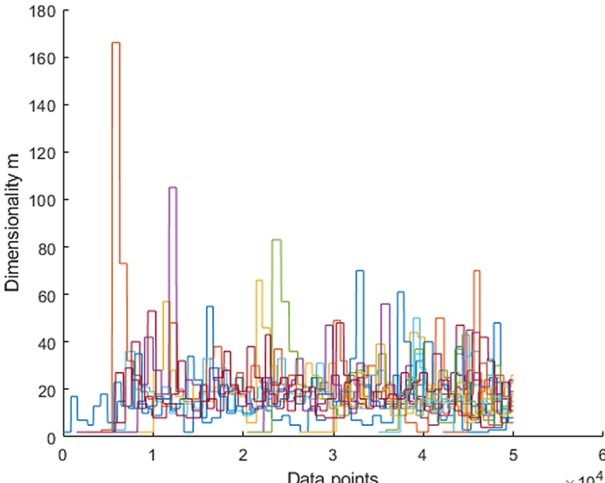

**Fig 11**. **Dimensionality (a) and number of units (b) time courses corresponding to the clustering on the h1024 data set in Fig 10.** (a) The average ground truth dimensionality of all clusters is $m = 15.5$ and the average dimensionality of all PCA units is $m = 16.7$ at the end of the training procedure.

measures were used as quality measures for all data sets. In addition, the number of clusters was used as an additional measure for algorithms with an adaptive number of units.

The CI values averaged over 26 runs, with the number of runs obtained from a power analysis (22), are presented in Table 1. The values are normalized by the number of clusters in each data set, with zero indicating a perfect CI value. The table contains two categories of algorithms. The first category comprises algorithms with a fixed number of units. These algorithms appear with bold font in the table. The second category are algorithms with an adaptive number of clusters. These appear with normal font. The table is sorted based on the last column, which shows the CI average across all data sets. As expected, the algorithms with a fixed number of units (in our tests this coincides with the true number clusters, which is, however, usually unknown) perform noticeably better than algorithms with an adaptive number of units. For the algorithms with an adaptive number of units, however our algorithm is ahead of the competition. It should be noted here that all other algorithms with adaptive unit numbers consider all data offline at once, whereas we update the model in an online setting, data point by data point. In addition, our adaptive dimensionality adjustment means that we can map each subspace with an optimal number of dimensions and thus save a lot of computational effort.

The results are further validated by looking at the NMI results in Table 2, with the best possible NMI being equal to one. While the algorithms with a fixed number of units are generally at the top, both our H-NGPCA algorithm and the Birch algorithm are able to outperform the Cure algorithm (with fixed number of units). This can certainly be seen as a success for both algorithms, as it means that the data is mapped better despite the adaptive number of units.

To corroborate our results, we show that the performance differences in NMI and CI between the 13 benchmarked methods are statistically significant. Because the preconditions for ANOVA are not met (no equal variances, no normally distributed residual values), we resort to the nonparametric Kruskal-Wallis test. For each of the measures NMI and CI, a single Kruskal-Wallis test is carried out. Overall, the sample size per method amounts to 416 (16 data sets with 26 random seeds/training runs each) and the degrees of freedom to 12. The null hypothesis can be rejected for all measures with $p < .001$ (two-sided) ($H = 2031.9$ for NMI, $H = 2411.0$ for CI; $H$ values are adjusted for tied ranks).

Furthermore, for post-hoc analysis, all possible 78 pairwise combinations between methods are subject to the Dunn test [30] with Bonferroni correction. Dunn's test statistic is a z-score. To achieve a significant result in a single pairwise comparison at $\alpha = 5\%$ (two-tailed; with Bonferroni correction), the absolute value of the z-score has to be greater than

**Table 1. Normalized CI results.** Bold method names indicate algorithms with a fixed number of clusters, otherwise the algorithm adaptivly determines the number of clusters. The last column shows the mean across all data sets. Results are sorted according to the last column. The corresponding standard deviations are shown in Table 9.

| | a1 | a2 | a3 | b1 | b2 | h32 | h64 | h128 | h256 | h512 | h1024 | s1 | s2 | s3 | s4 | u1 | mean |
|---|---|---|---|---|---|---|---|---|---|---|---|---|---|---|---|---|---|
| **C-NGPCA** | 0.01 | 0.01 | 0.03 | 0.00 | 0.00 | 0.00 | 0.01 | 0.00 | 0.01 | 0.03 | 0.09 | 0.00 | 0.00 | 0.00 | 0.00 | 0.50 | 0.04 |
| **Ward** | 0.00 | 0.00 | 0.00 | 0.00 | 0.09 | 0.00 | 0.00 | 0.00 | 0.00 | 0.00 | 0.00 | 0.00 | 0.00 | 0.00 | 0.00 | 0.62 | 0.04 |
| **MiniBatchKMeans++** | 0.04 | 0.03 | 0.03 | 0.05 | 0.11 | 0.00 | 0.00 | 0.00 | 0.00 | 0.00 | 0.00 | 0.01 | 0.03 | 0.04 | 0.04 | 0.55 | 0.06 |
| **GaussianMixture** | 0.04 | 0.03 | 0.03 | 0.03 | 0.09 | 0.00 | 0.00 | 0.00 | 0.00 | 0.00 | 0.00 | 0.00 | 0.04 | 0.04 | 0.04 | 0.62 | 0.06 |
| **SpectralClustering** | 0.00 | 0.02 | 0.00 | 0.01 | 0.13 | 0.00 | 0.07 | 0.13 | 0.06 | 0.03 | 0.18 | 0.00 | 0.00 | 0.00 | 0.06 | 0.48 | 0.07 |
| **AgglomerativeClustering** | 0.00 | 0.03 | 0.04 | 0.42 | 0.38 | 0.00 | 0.00 | 0.00 | 0.00 | 0.00 | 0.00 | 0.00 | 0.06 | 0.44 | 0.38 | 0.75 | 0.16 |
| **Cure** | 0.15 | 0.11 | 0.14 | 0.20 | 0.21 | 0.00 | 0.00 | 0.00 | 0.00 | 0.00 | 0.00 | 0.06 | 0.19 | 0.44 | 0.38 | 0.75 | 0.16 |
| H-NGPCA | 0.35 | 0.62 | 0.47 | 0.47 | 0.46 | 0.00 | 0.07 | 0.3 | 0.41 | 0.02 | 0.08 | 0.20 | 0.26 | 0.32 | 0.27 | 0.24 | 0.26 |
| Birch | 0.50 | 0.66 | 0.72 | 0.89 | 0.90 | 0.00 | 0.20 | 0.00 | 0.00 | 0.33 | 0.07 | 0.12 | 0.12 | 0.06 | 0.25 | 1.38 | 0.39 |
| AffinityPropagation | 0.70 | 0.80 | 0.80 | 1.00 | 1.00 | 0.00 | 0.07 | 0.00 | 0.00 | 0.00 | 0.00 | 0.44 | 0.38 | 0.44 | 0.44 | 0.62 | 0.42 |
| MeanShift | 0.20 | 0.57 | 0.72 | 0.81 | 0.83 | 1.00 | 1.00 | 1.00 | 1.00 | 1.00 | 1.00 | 0.00 | 0.00 | 0.00 | 0.12 | 1.12 | 0.65 |
| Clique | 0.55 | 0.63 | 0.76 | 0.99 | 0.96 | 1.00 | 1.00 | 1.00 | 1.00 | 1.00 | 1.00 | 0.31 | 0.38 | 0.81 | 0.81 | 0.62 | 0.80 |
| DBSCAN | 1.45 | 0.97 | 0.98 | 0.99 | 0.99 | 1.00 | 1.00 | 1.00 | 1.00 | 1.00 | 1.00 | 1.50 | 3.00 | 5.81 | 7.44 | 0.88 | 1.88 |

**Table 2. Normalized NMI results.** Bold method names indicate algorithms with a fixed number of clusters, otherwise the algorithm adaptivly determines the number of clusters. The last column shows the mean across all data sets. Results are sorted according to the last column. The corresponding standard deviations are shown in Table 10.

| | a1 | a2 | a3 | b1 | b2 | h32 | h64 | h128 | h256 | h512 | h1024 | s1 | s2 | s3 | s4 | u1 | mean |
|---|---|---|---|---|---|---|---|---|---|---|---|---|---|---|---|---|---|
| **GaussianMixture** | 0.97 | 0.98 | 0.98 | 0.97 | 0.97 | 1.00 | 1.00 | 1.00 | 1.00 | 1.00 | 1.00 | 0.99 | 0.94 | 0.79 | 0.73 | 0.98 | 0.96 |
| **Ward** | 0.94 | 0.97 | 0.98 | 0.92 | 0.92 | 1.00 | 1.00 | 1.00 | 1.00 | 1.00 | 1.00 | 0.98 | 0.93 | 0.78 | 0.71 | 1.00 | 0.95 |
| **MiniBatchKMeans++** | 0.93 | 0.97 | 0.99 | 0.95 | 0.92 | 1.00 | 1.00 | 1.00 | 1.00 | 1.00 | 1.00 | 0.98 | 0.94 | 0.78 | 0.71 | 0.96 | 0.95 |
| **SpectralClustering** | 0.96 | 0.98 | 0.99 | 0.97 | 0.92 | 1.00 | 0.96 | 0.93 | 0.98 | 0.99 | 0.85 | 0.98 | 0.94 | 0.80 | 0.72 | 0.69 | 0.92 |
| **C-NGPCA** | 0.96 | 0.97 | 0.97 | 0.95 | 0.99 | 0.98 | 0.95 | 0.93 | 0.92 | 0.91 | 0.90 | 0.99 | 0.94 | 0.79 | 0.73 | 0.74 | 0.91 |
| **AgglomerativeClustering** | 0.96 | 0.97 | 0.97 | 0.64 | 0.89 | 1.00 | 1.00 | 1.00 | 1.00 | 1.00 | 1.00 | 0.98 | 0.92 | 0.54 | 0.41 | 0.82 | 0.88 |
| H-NGPCA | 0.93 | 0.94 | 0.91 | 0.56 | 0.93 | 0.96 | 0.94 | 0.95 | 0.96 | 0.95 | 0.95 | 0.96 | 0.87 | 0.63 | 0.66 | 0.83 | 0.87 |
| Birch | 0.77 | 0.75 | 0.76 | 0.63 | 0.64 | 1.00 | 1.00 | 1.00 | 1.00 | 1.00 | 1.00 | 0.93 | 0.89 | 0.78 | 0.69 | 0.98 | 0.86 |
| **Cure** | 0.85 | 0.91 | 0.93 | 0.50 | 0.84 | 1.00 | 1.00 | 1.00 | 1.00 | 1.00 | 1.00 | 0.82 | 0.89 | 0.75 | 0.75 | 0.49 | 0.86 |
| AffinityPropagation | 0.68 | 0.67 | 0.69 | 0.00 | 0.00 | 1.00 | 0.98 | 1.00 | 1.00 | 1.00 | 1.00 | 0.84 | 0.80 | 0.70 | 0.64 | 0.98 | 0.75 |
| MeanShift | 0.86 | 0.80 | 0.76 | 0.69 | 0.72 | 0.00 | 0.00 | 0.00 | 0.00 | 0.00 | 0.00 | 0.99 | 0.95 | 0.80 | 0.72 | 0.99 | 0.52 |
| DBSCAN | 0.77 | 0.86 | 0.89 | 0.00 | 0.00 | 0.01 | 0.08 | 0.00 | 0.16 | 0.00 | 0.08 | 0.86 | 0.75 | 0.58 | 0.54 | 0.94 | 0.41 |
| Clique | 0.71 | 0.72 | 0.60 | 0.00 | 0.45 | 0.00 | 0.00 | 0.00 | 0.00 | 0.00 | 0.00 | 0.82 | 0.79 | 0.34 | 0.19 | 0.79 | 0.34 |

3.529. For NMI, this is the case for 62 out of 78 pairwise comparisons, for CI for 66 out of 78. Taking especially H-NCPGA into account, the performance of this algorithm is significantly different to all other algorithms except for affinity propagation in both NMI ($z = 2.972$, $p \approx .231$) and CI ($z = .676$, $p \approx 1.0$) and to BIRCH in CI ($z = .892$, $p \approx 1.0$). Nevertheless, it should be noted that both affinity propagation and BIRCH are offline algorithms which process all data at once, thus they are (i) not applicable to data streams and (ii) computationally heavy on large data sets. In contrast, H-NGPCA can be applied in an online manner, which constitutes a key advantage in handling dynamic or continuously evolving data.

Because we wanted to reject the null hypothesis for most pairwise comparisons in the post-hoc Dunn test, the power analysis is based on this test. Our goal was to keep the type II error below $\beta = 5\%$ for $\alpha = 5\%$ (two-tailed; with Bonferroni correction, resulting in $\tilde{\alpha} = 5\%/78$) and an effect size $\delta = 0.25$ (small to medium effect). Since Dunn's test statistic is a z-score, the required sample size is computed by

$$s = \left( \frac{z_\beta - z_{1-\tilde{\alpha}/2}}{\delta} \right)^2, \tag{22}$$

yielding $s \approx 409.4$ per method [31]. With a total of 16 data sets to be tested per method, this results after rounding in 26 training runs for each combination of the 16 data sets with the 13 methods.

## 6 Discussion and conclusions

Clustering data streams is difficult because the effectiveness of clustering algorithms depends on the correct assignment of hyperparameters. These parameters, which include the number of clusters, density thresholds, decay rates and window lengths, have a significant impact on the quality of the clustering results. In comparison to conventional clustering scenarios, data streams have a dynamic nature and are subject to continuous changes. Consequently, setting fixed hyperparameter values without prior insight risks skewing the clustering model. Among these parameters, the optimal number of clusters is of particular importance, and at least that requires an adaptive component.

Previously published work on the topic of continuously finding the optimal number of units rarely provided repositories to reproduce the results, which makes them unsuitable for benchmark. Also, most algorithms, when trying to remove the hyperparameter of the unit count, added more new hyperparameters in the process. This negates the added value of these methods at the very beginning.

Our algorithm presented in this work extends local PCA clustering by a hierarchical approach to continuously adapt the number of units based on a hyperparameter-free quality measure. We evaluated the performance of the presented algorithm in an experimental study on all data sets of the clustering benchmark database. The visual results showed successful training of the H-NGPCA algorithm on data sets with different characteristics, such as cluster overlap, unbalanced clusters and high dimensionality.

We propose a parameter-free splitting criterion based on intra- and inter-cluster distances that is weighted by each units activity. The quality measure rewards it if the data points that belong to a unit are close to it and if the data points that do not belong to the unit are far away.

For high-dimensional data we proposed an adaptive dimensionality control algorithm that controls each units dimensionality individually. This local dimensionality control is necessary because units require a different dimensionality depending on the data which they represent. For example, units that are at a higher hierarchical level often lie between several clusters rather than directly on one. This means that the variance of the unit no longer depends on the variance of the clusters but on the distances between them.

Further, we compared our algorithm with state-of-the-art clustering algorithms. First, we conducted an extensive literature review to identify the most popular online clustering algorithms. In the next step, we looked for the major potential competitors for our algorithm. Unfortunately, this ended with the realization that none of the direct competitors had provided code repositories which could be used to apply the algorithms in a comparison. We have therefore also extended the comparison to traditional offline clustering methods with a static or adaptive number of units. Compared to these, we naturally have a strong handicap due to online learning and the adaptive number of units. In this comparison, we scored positively based on the NMI and CI values, we are better than all other offline algorithms with an adaptive number of units and in some cases can also keep up with offline algorithms with a static number of units. Our estimated number of units is also closer to the actual number of units compared to the competing algorithms.

A total of five hyperparameters are used in H-NGPCA. One low-pass parameter for updating the low-passes, and two parameters each for dimensionality control and the number-of-unit algorithm. For the dimensionality control, it is necessary to specify how much variance is to be retained for each unit and how many training cycles the unit-specific dimensionality control pauses after a dimensionality change. For the split algorithm, there is a similar hyperparameter that prevents further splits after a recent split and a hysteresis parameter to prevent splits caused by statistical outliers. All these parameters can be set intuitively and do not require any complex tuning. Compared to other algorithms with an adaptive unit number, two hyperparameters are also used on average.

 

While our algorithm shows very convincing results overall, our algorithm has two weakpoints which we want to address in future work. Firstly, the concept of parent units with two children has the disadvantage that if three units lie along an axis, there is no split. This has already been shown and discussed in Fig 12. One approach could be to replace the strict binary (two children) structure with one that allows more than two children in these situations. Second, a merge mechanism is currently missing, where parts of the unit tree can be pruned again. However, it is possible to implement this with relatively little effort, as it is only necessary to regularly check whether the higher hierarchical levels are still relevant or not. Nevertheless, testing this component would exceed the scope of this work. For future work, we aim to further explore the application of H-NGPCA in real-world scenarios, particularly in biological and industrial domains [32]. These fields often involve complex and high-dimensional data streams, providing an ideal environment to assess the performance of the proposed method.

## List of symbols

| Roman | | | |
|---|---|---|---|
| **Symbol** | **Description** | **Symbol** | **Description** |
| $\mathbf{c}$ | PCA center | $d$ | distance measure |
| $H$ | entropy | $\mathbf{d}$ | vector of distances |
| $i$ | dimensionality index | $j$ | PCA unit index |
| $\mathbf{L}$ | cluster label | $m$ | number of eigenvectors |
| $M$ | number of local PCA units | $n$ | dimensionality of data space |
| $N$ | number of data points | $R$ | unit ranking |
| $\mathbf{R}$ | ranking vector | $t$ | time step |
| $V$ | unit volume | $\mathbf{W}$ | matrix of estimated eigenvectors |
| $\mathbf{w}_i$ | $i^{\text{th}}$ eigenvector estimate | $\mathbf{x}$ | vector drawn from data space |
| $\mathbf{y}$ | neuron activation | s | sample size |
| $\mathcal{L}$ | set of winner units | $\mathcal{L}$ | set of looser units |
| $\mathcal{U}_b$ | set of outermost dev. units | $\mathcal{U}_u$ | set of unborn units |
| $\mathcal{U}_d$ | set of dev. units | | |
| Greek | | | |
| $\alpha$ | type I error | $\beta$ | type II error |
| $\epsilon$ | learning rate | $\xi$ | distance between data and center |
| $\rho$ | neighborhood range | $\mu$ | low-pass filter parameter |
| $\check{\lambda}$ | mean residual variance | $\lambda$ | trained eigenvalue |
| $\Gamma$ | normalized matching measure | $\mathbf{\Lambda}$ | diagonal matrix of eigenvalues |
| $\mathcal{S}$ | data stream | $\gamma$ | dim. adj. delay |
| $\psi$ | hysteresis parameter | $\delta$ | effect size |
| $\sigma^2$ | residual variance | $\zeta$ | split adjustment delay |

## Appendix

## A Further details on related work

A complete literature review of data stream clustering algorithms is shown in Table 3. In particular interesting for the benchmarking are the following algorithms: ODAC [33], Adaptive K-means [34], StrAP [35], FEAC-Stream [36], Incremental DBSCAN [37], LDBSCAN [38], GCHDS [39], GSCDS [40] and CluDistream [41]. Unfortunately, reviewing

**Table 3. The properties that each clustering algorithm studied satisfies with respect to clustering problems on data streams.** '+' means that the property is present, '(+)' partly present, '-' not present, 'N.I.' no information available, and 'req' required. In the first three rows, the properties of our NGPCA versions are listed. Then, the best known algorithms from the five cluster categories are mentioned.

| Algorithm | Ref. | Adapt. Clust.# | Adaptive Parameter | High- dim. | Robust | Drift | Expert- know. | Fully Online |
|---|---|---|---|---|---|---|---|---|
| **Our algorithms** | | | | | | | | |
| NGPCA | [19] | – | - | + | – | - | req. | + |
| C-NGPCA | [22] | – | (+) | + | + | + | req. | + |
| H-NGPCA | | + | + | + | + | + | No | + |
| **Hierarchical** | | | | | | | | |
| Birch | [42] | + | – | - | + | (+) | req. | – |
| CURE | [43] | + | – | - | (+) | N.I. | req. | (+) |
| ODAC | [33] | + | – | (+) | – | + | req. | (+) |
| CHAMELEON | [44] | (+) | – | + | + | N.I. | req. | - |
| E-Stream | [45] | (+) | – | - | – | + | req. | + |
| HUE-Stream | [46] | (+) | – | - | + | + | req. | N.I. |
| **Partitioning** | | | | | | | req. | |
| Lsearch | [47] | – | (+) | – | - | – | req. | (+) |
| Inc. K-means | [48] | + | (+) | – | (+) | N.I. | req. | + |
| Adap. K-means | [34] | (+) | – | (+) | – | + | req. | + |
| Stream KM++ | [11] | – | (+) | – | - | – | No | + |
| StrAP | [35] | + | (+) | – | + | + | No | + |
| FEAC-Stream | [36] | + | (+) | – | + | + | req. | + |
| CLARA | [49] | – | - | N.I. | + | N.I. | req. | - |
| CluStream | [50] | – | - | – | - | + | req. | - |
| HPStream | [51] | – | - | + | – | + | req. | - |
| SWClustering | [52] | – | - | – | - | + | req. | + |
| **Density** | | | | | | | | |
| Inc. DBSCAN | [37] | + | – | (+) | + | N.I. | req. | + |
| DenStream | [53] | + | – | - | (+) | + | req. | - |
| rDenStream | [54] | + | – | (+) | + | N.I. | req. | - |
| LDBSCAN | [38] | + | – | (+) | + | N.I. | req. | + |
| SOStream | [55] | – | - | – | - | – | req. | - |
| MuDi-Stream | [56] | – | - | – | + | + | req. | - |
| CEDAS | [57] | – | - | + | + | + | req. | + |
| I-HASTREAM | [58] | + | (+) | + | + | (+) | No | - |
| OPCluStream | [59] | – | - | – | + | (+) | req. | + |
| OPTICS | [60] | – | - | – | + | + | req. | + |
| D-Stream | [13] | + | – | - | + | + | req. | - |
| MR-Stream | [14] | + | – | - | + | + | req. | (+) |
| DSCLU | [61] | (+) | – | - | N.I. | N.I. | req. | - |
| **Grid** | | | | | | | | |
| CLIQUE | [62] | + | – | (+) | (+) | – | No | (+) |
| WaveCluster | [63] | + | – | + | + | – | req. | N.I. |
| STING | [64] | + | + | – | + | + | No | - |
| DENGRIS | [65] | + | – | N.I. | + | + | req. | - |
| GCHDS | [39] | + | – | + | + | (+) | req. | + |
| GSCDS | [40] | + | – | + | (+) | N.I. | req. | + |
| DGClust | [66] | – | - | – | + | – | No | + |
| **Model** | | | | | | | | |
| CluDistream | [41] | + | – | N.I. | + | (+) | req. | + |
| SWEM | [67] | + | – | - | + | – | req. | - |
| COBWEB | [68] | + | + | – | + | – | req. | - |

these algorithms revealed that each publication (i) uses completely different data sets, most of which are simple two-dimensional offline data sets; (ii) does rarely specify the sampling order of the data sets; (iii) does not specify hyperparameters or training time; (iv) if benchmarks are available, these are against algorithms with completely different properties and (v) in most cases no working Github or other code-versioning repositories are available. There is a repository for

the algorithm ODAC [33], but only an offline implementation is available. For LDBSCAN [38] a repository exist, but there are no readme or docs that are necessary for use. The only well prepared repository is provided for the algorithm Inc. DBSCAN [37], whereby the owner of the repository describes that some parts of the algorithm are not described in the original paper and that the repository owner has implemented its own solutions for those algorithmic holes. No other competing algorithm provides an implementation of their algorithm. It is therefore not possible to carry out meaningful benchmarks with these algorithms, which is why we also had to consider eleven offline algorithms for the benchmark, such as Gaussian Mixture Models, K-Means, Birch and DBSCAN.

## B Pseudo-code of training process

The algorithm 2 shows the algorithmic steps necessary to re-implement the algorithm proposed in this work.

**Algorithm 2 H-NGPCA training procedure.** The dimensionality adjustment is presented in separate algorithm and the structure adaptation components in a separate figure.

```
1:  [c₀,Λ₀,W₀,ε₀,a₀,π₀,γ₀,σ₀,m₀,d̄_intra,0,d̄_inter,0] ← Init root unit
2:  [c_c,Λ_c,W_c,ε_c,a_c,π_c,γ_c,σ_c,m_c,d̄_intra,c,d̄_inter,c] ← Init root children, c = 1,2
3:  for all x ∈ S do                              ▷ Input vector from data stream
4:      L = U                                     ▷ Initial set of loser units
5:      W = {}                                    ▷ Initial set of winner units
6:      k = 0                                     ▷ Start at root unit
7:      do
8:          c₁,c₂ ← children of k
9:          d_c₁ ← Potential function (x, c_c₁,Λ_c₁,W_c₁)            ▷ (5)
10:         d_c₂ ← Potential function (x, c_c₂,Λ_c₂,W_c₂)
11:         winner child c_w ← arg min(d_c₁,d_c₂)                   ▷ (6)
12:         ā_c₁ ← Update assignment value (ā_c₁)               ▷ (8)–(10)
13:         ā_c₂ ← Update assignment value (ā_c₂)               ▷ (8)–(10)
14:         W ← W ∪ c_w
15:         L ← L \ c_w
16:         k ← c_w
17:     while Winner child c_w has children                      ▷ Descent in tree
18:     for all j ∈ W  do
19:         ε_j ← Adaptive learning rate (Λ_j,y_j,μ)        ▷ y_j(ξ) = W^T ξ_j
20:         c_j ← Update center (ε_j,c_j,x)
21:         [Λ_j,W_j] ← Online PCA (Λ_j,W_j)
22:         σ²_j ← update residual variance (σ²_j,ε_j,ξ_j,y_j)     ▷ Only for m < n
23:         [m_j, Λ_j,W_j,σ²_j] ← Dim. adjustment (m_j, Λ_j,W_j,σ²_j)   ▷ Alg. 1
24:         d̄_intra,j ← Update intra-cluster distance with d_j    ▷ d_j set in line 9 and 10
25:     end for
26:     for all j ∈ L do
27:         d̄_inter,j ← Update inter-cluster distance with d_j    ▷ d_j set in line 9 and 10
28:     end for
29:     for all j ∈ U_b do
30:         π̄_j ← Update activity (π̄_j)                        ▷ (12)–(14)
31:     end for
32:     Unit tree ← Structure adaptation (d̄_intra,d̄_inter)       ▷ Fig 7
33: end for
```

## C Details of benchmark data sets and evaluation metrics

The data sets considered for the benchmark are shown in Table 4. They vary in the number of clusters, data points per cluster, cluster overlap, dimensionality, and the data point balance. The high-dimensional data sets are particularly interesting, since it is difficult to find the right number of clusters in high-dimensional spaces, as they are usually only sparsely filled.

The Centroid Index (CI) is a metric derived from the cluster centers to evaluate the cluster-level mismatch [69]. It compares the local PCA centers $c_j$ with the true cluster centres $c_i'$. For each cluster center $c_i'$, the nearest PCA unit $c_j$ is determined using the Mahalanobis distance (4) with respect to the clusters eigenvalues and eigenvectors. Eigenvalues and eigenvectors are computed for each cluster based on their respective covariance matrices. PCA units without matches are labeled as orphans or "dead". The measure $CI(c', c)$ is then the sum of all orphan units. It's important to note that the CI is asymmetric ($c \rightarrow c' \neq c' \rightarrow c$) [69]. Therefore, $CI(c, c')$ is calculated similarly, matching clusters to local PCA units, using the eigenvalues and eigenvectors of the PCA units. The symmetric version $CI_2$ [69] used in this work is obtained by

$$CI_2(c, c') = \max\left(CI(c, c'), CI(c', c)\right). \tag{23}$$

With the symmetric variant ($CI_2(c, c')$), the number of clusters does not matter because the cluster index $CI_2$ is not limited by the pairing as other set-based measures. Instead, it gives a value that is equal to the difference between the number of clusters and number of units, or higher if other cluster-level mismatches are detected; if no orphans exist, each PCA unit is mapped to exactly one cluster indicating that the structures are close to each other ($CI_2 = 0$). In the following, CI stands for $CI_2$ and is in addition normalized by the number of clusters in a data set.

The Normalized Mutual Information (NMI) is an external measure expressing how much information is shared between the real and the predicted clustering. Therefore, ground truth information about the real cluster assignment is required for this measure. For the benchmark data sets in [29] on which we rely, this information is provided. For each data point $i$, the ground truth cluster label $l_i$ is set to the corresponding cluster index, varying between 1 and the number of clusters $M$. $L = \{l_i\}_{i=1}^{N}$ is the set of all ground truth labels. The predicted cluster labels $L' = \{l_i'\}_{i=1}^{N}$ are obtained by determining which of the local PCA units is closest to the respective data point. The distance measure is the Mahalanobis distance (4).

**Table 4. Clustering benchmark data sets [29].**

| Data set | Data points $N$ | Clusters $M$ | Overlap | Unbalanced | Dim. $n$ |
|---|---|---|---|---|---|
| S1 | 5000 | 15 | No | No | 2 |
| S2 | 5000 | 15 | No | No | 2 |
| S3 | 5000 | 15 | Yes | No | 2 |
| S4 | 5000 | 15 | Yes | No | 2 |
| A1 | 3000 | 20 | No | No | 2 |
| A2 | 5250 | 35 | No | No | 2 |
| A3 | 7500 | 50 | No | No | 2 |
| B1 | 100000 | 100 | Yes | No | 2 |
| B2 | 100000 | 100 | No | No | 2 |
| U1 | 6500 | 8 | No | Yes | 2 |
| h32 | 1024 | 16 | Yes | No | 32 |
| h64 | 1024 | 16 | No | No | 64 |
| h128 | 1024 | 16 | No | No | 128 |
| h256 | 1024 | 16 | No | No | 256 |
| h512 | 1024 | 16 | No | No | 512 |
| h1024 | 1024 | 16 | No | No | 1024 |

The NMI is a normalized form of the mutual information (MI). In the literature, several versions of the NMI are proposed. We use the following definition [70]:

$$\text{NMI}(\mathbf{L}', \mathbf{L}) = \frac{\text{MI}(\mathbf{L}', \mathbf{L})}{\sqrt{H(\mathbf{L}')H(\mathbf{L})}} \tag{24}$$

with $\text{MI}(\mathbf{L}', \mathbf{L}) = H(\mathbf{L}) + H(\mathbf{L}') - H(\mathbf{L}', \mathbf{L})$, $H(\mathbf{L}) = -\sum_{j=1}^{N} p_j \log_2(p_j)$ (entropy with cluster label probabilities $p_j$) and $H(\mathbf{L}', \mathbf{L}) = -\sum_{i=1}^{N} \sum_{j=1}^{N} p_{i,j} \log_2(p_{i,j})$. The NMI expresses the amount of information the overall local PCA model can extract from the ground truth distribution. A value of 1.0 is the maximum and indicates that real and predicted clustering are identical.

## D Algorithm parameterizations

The hyperparameters shown below for each algorithm are those used to generate the results of Table 1, Table 2 and Table 7. We limit the parameter overview to those that vary from the default values. For a full list of parameters we refer to the corresponding scikit-learn and pyclustering documentation or to our benchmark-script within our GitHub repository. The N-clusters parameter is set when required to the number of clusters in the respective ground truth data set (Table 5).

Table 6 shows the parameters set for the benchmark results of our algorithm. The dimensionality $m_j$, not mentioned in the table, is initialized to two, and adaptively adjusted for each unit individually using the adaptive dimensionality control. The low-pass filter $\mu$, dimensionality threshold and dimensionality adjustment unit-specific delay $\gamma_j$ are fixed values obtained from [22,25]. The hysteresis parameter $\psi$ is chosen close to 1, with slightly smaller values for data sets with high cluster overlap (e.g. S3, S4). The delay after each splitting operation $\zeta$ varies depending on the number of samples in a data set.

**Table 5.** Parameters set for each algorithm to generate the results of Table 1, Table 2 and Table 7.

| Algorithm | Hyperparam. 1 | Hyperparam. 2 | Hyperparam. 3 |
|---|---|---|---|
| Cure | N-clusters | | |
| Clique | Interval = 20 | Threshold = 10 | |
| MB KMeans++ | N-clusters | | |
| Aff. Propagation | Damping = 0.95 | Preference = -200 | |
| MeanShift | Bandwidth = 0.05 | | |
| Spec. Clustering | N-clusters | Eigensolver = arpack | Affinity = near. neigh. |
| Ward | N-clusters | Linkage = ward | Connectivity = 10 |
| Agg. Clustering | N-clusters | Linkage = average | Metric = cityblock |
| DBSCAN | eps = 0.05 | | |
| Birch | | | |
| GMM | N-clusters | Cov. type = Full | |

**Table 6.** Parameters set for our H-NGPCA algorithm to generate the results of Table 1, Table 2 and Table 7.

| Initialization | | | |
|---|---|---|---|
| $\epsilon_j$ | 0.99 | $\mathbf{W}_j$ | random orthogonal vectors |
| $\Lambda_j$ | diagonal elements set to 1 | $\mathbf{c}_j$ | random data points |
| $d_{\text{intra},j}$ | 1 | $d_{\text{inter},j}$ | 1 |
| Hyperparameters | | | |
| $\mu$ | 0.005 | $\gamma_j$ | 50 |
| Dim. threshold | 0.7 | $\zeta$ | set between 50 and 250 |
| $\psi$ | set between 0.97 and 0.99 | | |

## E Additional experimental results and discussions

### E.1 Extended visual results

In Fig 12, the H-NGPCA algorithm is trained on the s2 data set. In contrast to the s1 data set, this data set stands out due to the overlap of the clusters. Our algorithm shows no weakness when training on strongly overlapping clusters and can also determine cluster centers and shapes in overlap regions. Nevertheless, one weakness of the algorithm becomes clear. If three clusters are in a straight line, the splitting approach used here may fail. This is because the parent unit lies appropriately on the middle cluster and extends along an axis to the two others. The corresponding children each lie between the center cluster and an outer cluster, which means that the fit of the parent unit is better than that of the child units. Therefore, no split is performed. The associated learning rate curve (Fig 13a) shows that the learning rates of all units quickly converge towards zero. The number of units (Fig 13b) increases steadily until it reaches the point where all units but the one covering three clusters are correctly split and then the number stays constant.

Further, H-NGPCA was tested on the u1 data set. This data set is characterized by strongly unbalanced clusters (with many data points in the three clusters on the left). Previous versions of NGPCA [22] had major problems on the data set. The H-NGPCA version presented here can learn this data set correctly without any problems. In Fig 14 the final clustering is shown with all clusters being represented by exactly one unit and a correct clustering of the data points. In Fig 15 the corresponding time courses of the quality measure and the number of units are shown. The quality measure starts high as the units are still adapting. As soon as the units converge, this also happens with the quality measure. New units that only become independent later in the training do not exhibit the same behavior, as the units learning rates have already reduced considerably by this time. The progression of the number of units looks similar to the previous s2 data set. The correct number of units is reached quickly and then remains stable.

### E.2 Extended quantitative results

We compared the real number of clusters with the number approximated by a clustering algorithm. As part of the benchmarked algorithms have a fixed number of units, we only performed that test for algorithms with an adaptive number. The

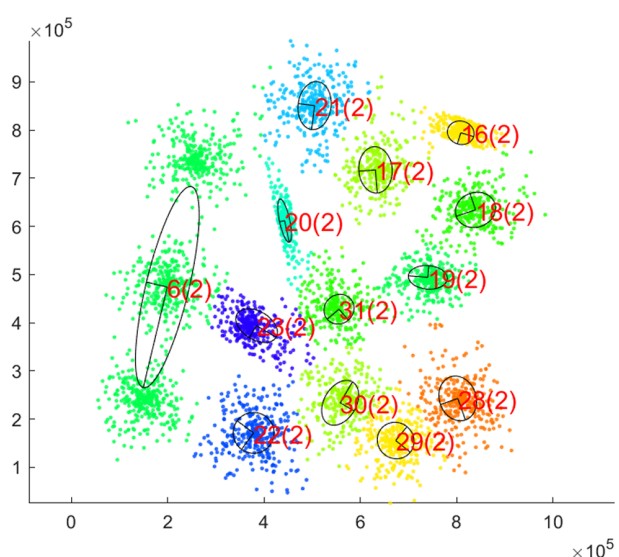

**Fig 12. Final clustering on the s2 data set. All PCA units are represented with an axis length of $\sqrt{\lambda_i}$.**

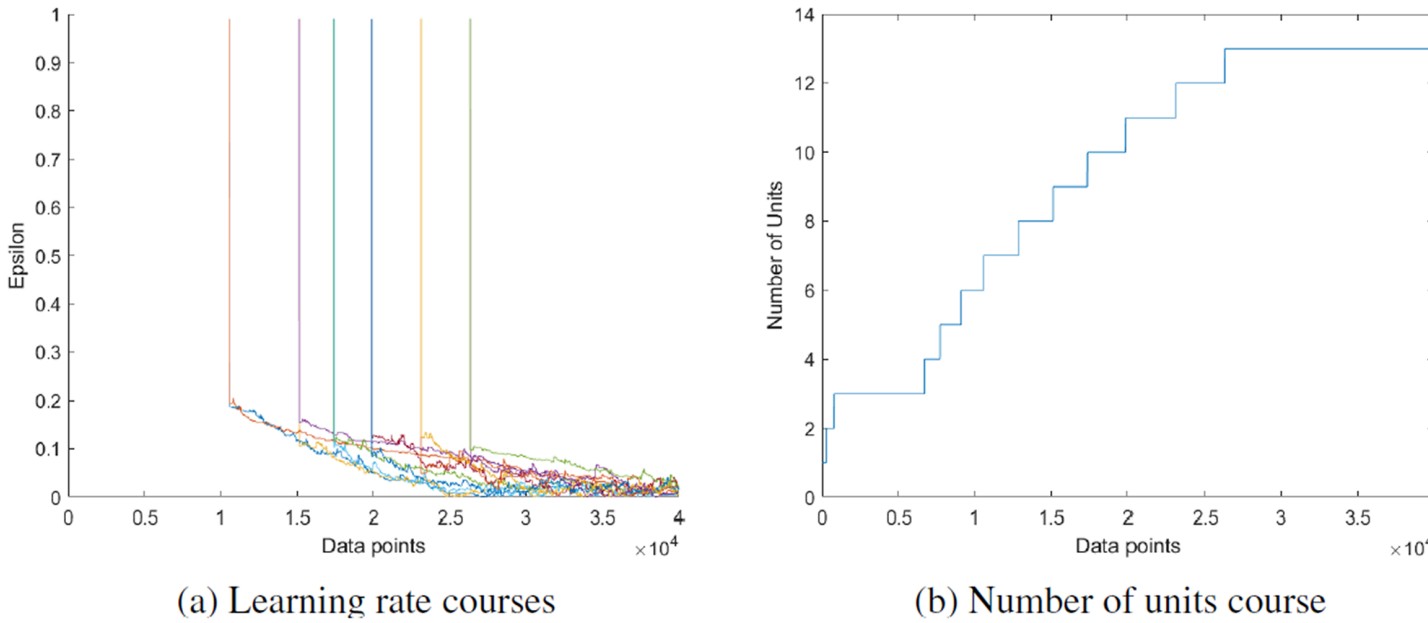

(a) Learning rate courses   (b) Number of units course

**Fig 13. Time courses of the learning rate and number of units corresponding to the clustering for data set s2 in Fig 12.**

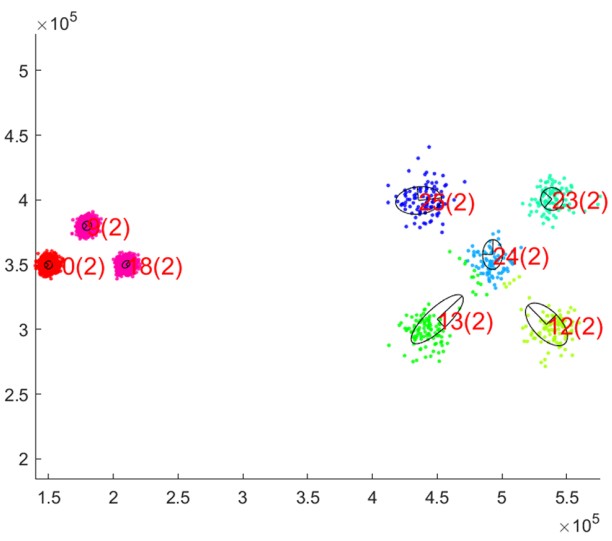

**Fig 14. Final clustering on the u1 data set. All PCA units are represented with an axis length of $\sqrt{\lambda_i}$.**

results are shown in Table 7. The majority of competing algorithms tend to significantly underestimate the number of clusters in a data set. This effect is particularly powerful on the b-series, as these data sets consist of 100 clusters. In general, most competing algorithms seem to stagnate somewhere around 15 clusters. We further discuss this in section 6 and it could be interesting to investigate this in a larger study. Our algorithm is always close to the real number of clusters regardless of the amount of clusters. Visual results of all competing algorithms are available in a compact form within the supplementary material. A final cluster result is shown there as an example for each combination of data set and algorithm. Empty cells mean that the algorithm could not be applied to the data set.

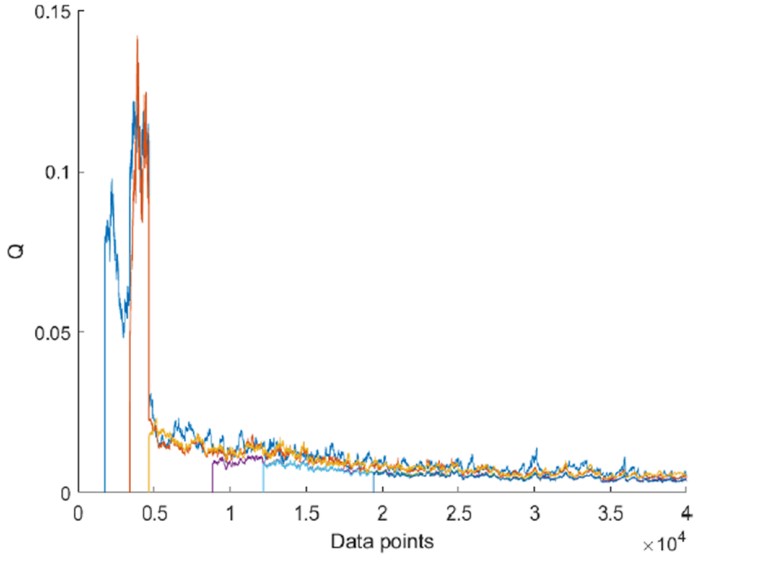

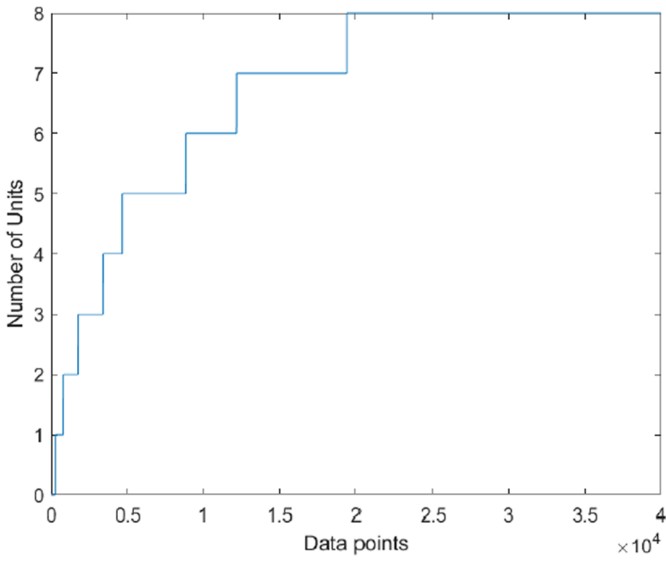

(a) Quality measure of unborn units

(b) Number of units course

**Fig 15**. Quality measure and number of units courses corresponding to the clustering on the u1 data set in Fig 14.

**Table 7**. Averaged number of units compared to the actual number of clusters for algorithms with an adaptive number of units.

| | a1 | a2 | a3 | b1 | b2 | h32 | h64 | h128 | h256 | h512 | h1024 | s1 | s2 | s3 | s4 | u1 |
|---|---|---|---|---|---|---|---|---|---|---|---|---|---|---|---|---|
| Ground Truth | 20 | 35 | 50 | 100 | 100 | 15 | 15 | 15 | 15 | 15 | 15 | 16 | 16 | 16 | 16 | 8 |
| Clique | 9.00 | 14.00 | 12.00 | 1.00 | 4.00 | 0.00 | 0.00 | 0.00 | 0.00 | 0.00 | 0.00 | 16.00 | 10.00 | 2.00 | 2.00 | 8.00 |
| AffinityPropagation | 6.00 | 7.00 | 10.00 | 0.00 | 0.00 | 16.00 | 15.00 | 16.00 | 16.00 | 16.00 | 16.00 | 8.00 | 9.00 | 8.00 | 8.00 | 6.00 |
| MeanShift | 16.00 | 15.00 | 14.00 | 19.00 | 17.00 | 0.00 | 0.00 | 0.00 | 0.00 | 0.00 | 0.00 | 15.00 | 15.00 | 15.00 | 17.00 | 14.00 |
| Birch | 10.00 | 12.00 | 14.00 | 11.00 | 10.00 | 16.00 | 19.00 | 16.00 | 16.00 | 21.00 | 17.00 | 13.00 | 14.00 | 14.00 | 18.00 | 16.00 |
| DBSCAN | 49.00 | 40.00 | 44.00 | 2.00 | 2.00 | 2.00 | 4.00 | 1.00 | 12.00 | 1.00 | 5.00 | 39.00 | 63.00 | 108.00 | 134.00 | 8.00 |
| H-NGPCA | 20.3 | 24.3 | 44 | 37 | 69 | 15.1 | 15.1 | 14.6 | 15.2 | 14.4 | 14.5 | 15 | 15 | 10 | 12 | 9.12 |

The standard deviations for both the CI and NMI results (Table 1-Table 2) are shown in Table 8 and Table 9. Some algorithms, in particular those with a static number of units, are fully deterministic and therefore have standard deviations of zero. On the other hand, some of the algorithms with a dynamic number of units are unsuitable for certain data sets. This also leads to a standard deviation of 0, as the algorithms are stuck with one single unit across all training runs.

In non-stationary and transient environments, data distributions may change over time. So far, only data distributions in which the number of clusters is constant have been considered. In the following, a situation is considered in which the distribution changes in the middle of training, so that the units have to readjust and the optimal number of units changes (Fig 16). The H-NGPCA model is initially trained on a square with a line attached (Fig 16(a)). After the model converged and all units' learning rates cooled down, the distribution is abruptly extended by a ring (Fig 16(b)). All units wake up again to adjust to the new distribution and split up whenever necessary. As our algorithm does not currently have a merge function which reduces the number of units, there is a risk that incorrect splits will occur when readjusting the units shortly after changing the data distribution, leading to dead units. A weaker effect can be seen in our case, where units 25 and 27 are not dead, but overlap unnecessarily. Ideally, these two units should be merged. Nevertheless, this small experiment shows that the H-NGPCA is suitable for data stream clustering.

**Table 8**. CI standard deviations corresponding to Table 1.

| | a1 | a2 | a3 | b1 | b2 | h32 | h64 | h128 | h256 | h512 | h1024 | s1 | s2 | s3 | s4 | u1 |
|---|---|---|---|---|---|---|---|---|---|---|---|---|---|---|---|---|
| AffinityPropagation | 0.00 | 0.00 | 0.00 | 0.00 | 0.00 | 0.00 | 0.00 | 0.00 | 0.00 | 0.00 | 0.00 | 0.00 | 0.00 | 0.00 | 0.00 | 0.00 |
| AgglomerativeClustering | 0.00 | 0.00 | 0.00 | 0.00 | 0.00 | 0.00 | 0.00 | 0.00 | 0.00 | 0.00 | 0.00 | 0.00 | 0.00 | 0.00 | 0.00 | 0.00 |
| Birch | 0.00 | 0.00 | 0.00 | 0.00 | 0.00 | 0.00 | 0.00 | 0.00 | 0.00 | 0.00 | 0.00 | 0.00 | 0.00 | 0.00 | 0.00 | 0.00 |
| C-NGPCA | 0.02 | 0.01 | 0.01 | 0.01 | 0.01 | 0.07 | 0.01 | 0.03 | 0.01 | 0.04 | 0.02 | 0.00 | 0.00 | 0.01 | 0.00 | 0.00 |
| Clique | 0.00 | 0.00 | 0.00 | 0.00 | 0.00 | 0.00 | 0.00 | 0.00 | 0.00 | 0.00 | 0.00 | 0.00 | 0.00 | 0.00 | 0.00 | 0.00 |
| Cure | 0.00 | 0.00 | 0.00 | 0.00 | 0.00 | 0.00 | 0.00 | 0.00 | 0.00 | 0.00 | 0.00 | 0.00 | 0.00 | 0.00 | 0.00 | 0.00 |
| DBSCAN | 0.00 | 0.00 | 0.00 | 0.00 | 0.00 | 0.00 | 0.00 | 0.00 | 0.00 | 0.00 | 0.00 | 0.00 | 0.00 | 0.00 | 0.00 | 0.00 |
| GaussianMixture | 0.60 | 0.75 | 0.70 | 0.85 | 1.67 | 0.00 | 0.00 | 0.00 | 0.00 | 0.00 | 0.00 | 0.47 | 0.56 | 0.49 | 0.58 | 0.47 |
| H-NGPCA | 0.17 | 0.11 | 0.23 | 0.22 | 0.07 | 0.00 | 0.11 | 0.06 | 0.08 | 0.05 | 0.07 | 0.11 | 0.13 | 0.10 | 0.09 | 0.18 |
| MeanShift | 0.00 | 0.00 | 0.00 | 0.00 | 0.00 | 0.00 | 0.00 | 0.00 | 0.00 | 0.00 | 0.00 | 0.00 | 0.00 | 0.00 | 0.00 | 0.00 |
| MiniBatchKMeans | 0.65 | 0.53 | 0.75 | 1.24 | 1.42 | 0.00 | 0.00 | 0.00 | 0.00 | 0.00 | 0.00 | 0.00 | 0.51 | 0.55 | 0.64 | 0.45 |
| SpectralClustering | 0.00 | 0.62 | 0.00 | 0.51 | 0.98 | 0.00 | 0.87 | 1.46 | 1.01 | 0.58 | 1.39 | 0.00 | 0.00 | 0.00 | 0.00 | 0.69 |
| Ward | 0.00 | 0.00 | 0.00 | 0.00 | 0.00 | 0.00 | 0.00 | 0.00 | 0.00 | 0.00 | 0.00 | 0.00 | 0.00 | 0.00 | 0.00 | 0.00 |

**Table 9**. NMI standard deviations corresponding to Table 2.

| | a1 | a2 | a3 | b1 | b2 | h32 | h64 | h128 | h256 | h512 | h1024 | s1 | s2 | s3 | s4 | u1 |
|---|---|---|---|---|---|---|---|---|---|---|---|---|---|---|---|---|
| AffinityPropagation | 0.00 | 0.00 | 0.00 | 0.00 | 0.00 | 0.00 | 0.00 | 0.00 | 0.00 | 0.00 | 0.00 | 0.00 | 0.00 | 0.00 | 0.00 | 0.00 |
| AgglomerativeClustering | 0.00 | 0.00 | 0.00 | 0.00 | 0.00 | 0.00 | 0.00 | 0.00 | 0.00 | 0.00 | 0.00 | 0.00 | 0.00 | 0.00 | 0.00 | 0.00 |
| Birch | 0.00 | 0.00 | 0.00 | 0.00 | 0.00 | 0.00 | 0.00 | 0.00 | 0.00 | 0.00 | 0.00 | 0.00 | 0.00 | 0.00 | 0.00 | 0.00 |
| C-NGPCA | 0.02 | 0.01 | 0.01 | 0.01 | 0.00 | 0.03 | 0.02 | 0.03 | 0.01 | 0.02 | 0.02 | 0.00 | 0.00 | 0.01 | 0.00 | 0.02 |
| Clique | 0.00 | 0.00 | 0.00 | 0.00 | 0.00 | 0.00 | 0.00 | 0.00 | 0.00 | 0.00 | 0.00 | 0.00 | 0.00 | 0.00 | 0.00 | 0.00 |
| Cure | 0.00 | 0.00 | 0.00 | 0.00 | 0.00 | 0.00 | 0.00 | 0.00 | 0.00 | 0.00 | 0.00 | 0.00 | 0.00 | 0.00 | 0.00 | 0.00 |
| DBSCAN | 0.00 | 0.00 | 0.00 | 0.00 | 0.00 | 0.00 | 0.00 | 0.00 | 0.00 | 0.00 | 0.00 | 0.00 | 0.00 | 0.00 | 0.00 | 0.00 |
| GaussianMixture | 0.02 | 0.01 | 0.00 | 0.00 | 0.01 | 0.00 | 0.00 | 0.00 | 0.00 | 0.00 | 0.00 | 0.01 | 0.01 | 0.01 | 0.00 | 0.04 |
| H-NGPCA | 0.08 | 0.07 | 0.14 | 0.10 | 0.02 | 0.04 | 0.04 | 0.05 | 0.03 | 0.03 | 0.04 | 0.08 | 0.08 | 0.06 | 0.04 | 0.09 |
| MeanShift | 0.00 | 0.00 | 0.00 | 0.00 | 0.00 | 0.00 | 0.00 | 0.00 | 0.00 | 0.00 | 0.00 | 0.00 | 0.00 | 0.00 | 0.00 | 0.00 |
| MiniBatchKMeans | 0.01 | 0.01 | 0.01 | 0.01 | 0.00 | 0.00 | 0.00 | 0.00 | 0.00 | 0.00 | 0.00 | 0.00 | 0.01 | 0.01 | 0.01 | 0.04 |
| SpectralClustering | 0.00 | 0.00 | 0.00 | 0.00 | 0.00 | 0.00 | 0.02 | 0.06 | 0.02 | 0.01 | 0.07 | 0.00 | 0.00 | 0.00 | 0.00 | 0.04 |
| Ward | 0.00 | 0.00 | 0.00 | 0.00 | 0.00 | 0.00 | 0.00 | 0.00 | 0.00 | 0.00 | 0.00 | 0.00 | 0.00 | 0.00 | 0.00 | 0.00 |

## 6.1 E.3 Per-Data-Point complexity analysis

Each data point traverses from the root to a leaf in a binary tree of depth $d$. Given the data dimensionality $n$ and PCA unit dimensionality $m$, the following algorithmic results for each data point presentation:

**Mahalanobis comparisons:** At each of the $d$–1 levels (excluding root level), the data point is compared to two child nodes using the Mahalanobis distance (5). Each comparison requires the projecting $x \in \mathbb{R}^n$ into PCA space using stored eigenvectors with $O(nm)$ and computing the Mahalanobis distance with diagonal covariance (from eigenvalues) with $O(m)$. Since two comparisons are made per level, the total cost across all levels is:

$$O((d-1) \cdot 2 \cdot (nm + m)) = O(nmd)$$

**PCA updates:** After choosing the winning child at each level, the algorithm updates the PCA parameters of all $d$ visited nodes (including the root). Each update costs according to [26]:

$$O(nm^2)$$

The total costs is multiplied by the depth $d$ in a sequential setting. As this step is performed in parallel in our setup, the cost remain $O(nm^2)$.

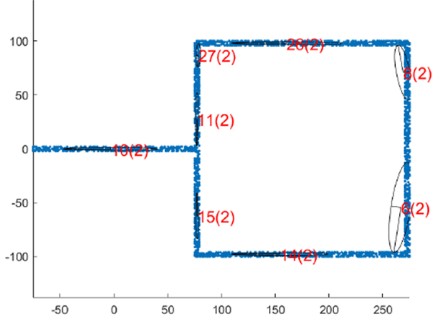

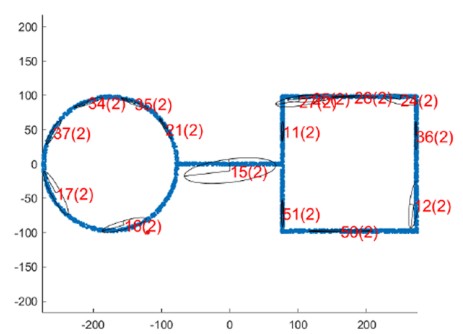

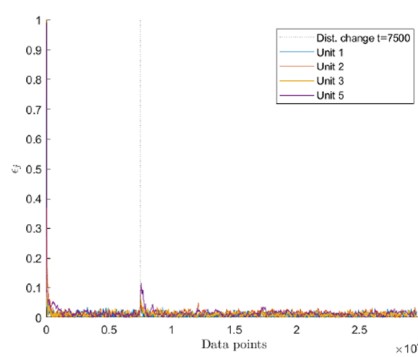

(a) Clustering on a line-square data set immediately before the distribution is extended.

(b) Final clustering on the ring-line-square distribution

(c) Corresponding learning rates of four units.

**Fig 16**. **The data sampling is extended from a line-square to a ring-line-square in the middle of the training.** The already converged network wakes up, and expands the tree to approximate the extended distribution. The corresponding learning rates show a hike when the data distribution is extended, but cools down quickly.

**Unit split:** The split decision are based on the already calculated Mahalanobis distances. For the direct comparison we loop over all candidates which has a complexity of

$$O(2^{d-1}).$$

**Dimensionality adjustment:** The dimensionality adjustment algorithm involves a linear eigenvalue regression and a Gram-Schmidt orthogonalization. The linear eigenvalue regression is only performed on the first $m$ eigenvalues which scales linearly with $O(m)$. The Gram-Schmidt orthogonalization is only performed when the dimensionality is increased. In the case when the PCA unit dimensionality $m$ is increased from an existing set of eigenvectors $m_{\text{old}}$, we only orthogonalize the newly added $\Delta m$ eigenvectors (Gram–Schmidt) against the already existing $m_{\text{old}}$. This leads to

$$O(nm_{\text{old}}\Delta m) \tag{25}$$

which is typically much smaller than a full recomputation $O(nm^2)$, especially when $\Delta m \ll m_{\text{old}}$. Therefore, the complexity of the dimensionality adjustment algorithm is omitted. Combining all terms, the overall per-data-point complexity is:

$$O(nmd + nm^2 + 2^{d-1}) = O(nm(d + m) + 2^{d-1})$$

The overall complexity grows linearly with $n$ and depends quadratically on $m$ which is usually much smaller $m \ll n$, plus the exponential term in $d$. The algorithm has therefore a higher computational complexity on data with many high-dimensional clusters.

# Author contributions

**Conceptualization:** Nico Migenda, Ralf Möller, Wolfram Schenck.

**Formal analysis:** Nico Migenda, Ralf Möller.

**Funding acquisition:** Wolfram Schenck.

**Methodology:** Nico Migenda, Ralf Möller, Wolfram Schenck.

**Project administration:** Ralf Möller.

**Resources:** Wolfram Schenck.

**Software:** Nico Migenda.

**Supervision:** Ralf Möller, Wolfram Schenck.

**Validation:** Nico Migenda, Ralf Möller, Wolfram Schenck.

**Visualization:** Nico Migenda.

**Writing – original draft:** Nico Migenda.

**Writing – review & editing:** Nico Migenda, Ralf Möller, Wolfram Schenck.

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
