## [Decision Letter · Decision Letter 0]

24 Sep 2025

PONE-D-25-43831H-NGPCA: Hierarchical clustering of data streams with adaptive number of clusters and adaptive dimensionalityPLOS ONE

Dear Dr. Migenda,

Thank you for submitting your manuscript to PLOS ONE. After careful consideration, we feel that it has merit but does not fully meet PLOS ONE’s publication criteria as it currently stands. Therefore, we invite you to submit a revised version of the manuscript that addresses the points raised during the review process.

We look forward to receiving your revised manuscript.

Kind regards,

Muhammad Ahsan, Ph.D.

Academic Editor

PLOS ONE

 [This research was funded by the German Federal Ministry of Education and Research (BMBF) in the project VIP4PAPS, grant number 03VP10031. The sole responsibility for the content of this publication lies with the authors.]. 

Additional Editor Comments (if provided):

Reviewer #1:

Reviewer #2:

Reviewer #3:

Reviewers' comments:

Reviewer's Responses to Questions

**Comments to the Author**

1. Is the manuscript technically sound, and do the data support the conclusions?

Reviewer #1: Yes

Reviewer #2: Yes

Reviewer #3: Yes

2. Has the statistical analysis been performed appropriately and rigorously?

Reviewer #1: N/A

Reviewer #2: Yes

Reviewer #3: Yes

3. Have the authors made all data underlying the findings in their manuscript fully available?

Reviewer #1: No

Reviewer #2: Yes

Reviewer #3: Yes

4. Is the manuscript presented in an intelligible fashion and written in standard English?

Reviewer #1: Yes

Reviewer #2: Yes

Reviewer #3: Yes

5. Review Comments to the Author

Reviewer #1: This manuscript introduces H-NGPCA, a hierarchical clustering algorithm for data streams that adaptively determines both the number of clusters and their local dimensionality. The method combines centroid-based (Neural Gas), model-based (PCA), and hierarchical clustering in a fully online manner. The work is technically sound, well-motivated, and addresses an important challenge in streaming data analysis. The experimental evaluation is thorough, and the visualizations effectively illustrate the algorithm’s behavior. However, several aspects require clarification and improvement to strengthen the contribution and ensure reproducibility.

The pseudo-code in Appendix B (Algorithm 2) is a valuable addition, but the main text does not sufficiently explain the flow of the algorithm. For example, the interaction between the tree traversal (line 6–16) and the update steps (line 17–27) is unclear. A high-level summary in Section 3.1 would help readers understand the end-to-end process before diving into details.

The authors rightly note that many online algorithms lack implementations, so they compare against offline methods. However, this puts H-NGPCA at a disadvantage. To fairness, consider including a streaming variant of a classic method (e.g., Streaming K-Means) even if not adaptive. In addition, I suggest that the author discuss more about the application of the method. https://doi.org/10.1016/j.compbiomed.2023.107244 DOI: 10.1109/ACCESS.2020.2970838

Table 1 and 2 show that H-NGPCA performs well, but the standard deviations are missing. Including these would help assess the stability of the results across multiple runs.

The claim that H-NGPCA outperforms BIRCH and Affinity Propagation in some cases is supported, but the discussion should emphasize that these are offline methods. The online capability of H-NGPCA is a significant advantage that should be highlighted more clearly.

The complexity analysis in Appendix E.3 is detailed and correct. The per-data-point complexity does not account for the dimensionality adjustment (Algorithm 1), which involves Gram-Schmidt and eigenvalue regression. This should be included in the analysis.

The authors acknowledge that H-NGPCA cannot handle three collinear clusters (Fig. 12) and lacks a merge mechanism. These are significant limitations. A brief discussion of how a merge mechanism could be integrated (e.g., via a quality measure pruning branches) would strengthen the paper.

The algorithm currently uses five hyperparameters. While this is fewer than many streaming algorithms, a table showing the values used for each dataset (or a justification for fixed values) would aid reproducibility.

Reviewer #2: The paper is well-structured and clearly written, presenting a timely and relevant study on H-NGPCA. The hierarchical extension is explained coherently and supported by appropriate experimental validation. Here are some suggestions to further enhance the paper.

Abstract

•The abstract could be strengthened by more explicitly highlighting the novelty of the proposed method. Additionally, presenting key quantitative results would give readers a clearer and more concrete understanding of the contribution and its significance.

Introduction

•The introduction clearly outlines the problem statement; however, the review of related work is somewhat limited. A more thorough discussion of existing dimensionality reduction techniques (e.g., Kernel PCA, Autoencoders, Deep Learning–based methods, t-SNE, UMAP) is recommended. Furthermore, the manuscript should more explicitly articulate the advancement of H-NGPCA over conventional NGPCA, providing a stronger justification for the incorporation of the hierarchical structure.

•A comparison of these dimensionality reduction techniques, along with a clear justification of their relevance, would strengthen the discussion.

•To further strengthen the problem statement, it is advisable to include supporting references that validate and contextualize the identified research gap.

Methodology

•The methodological description is detailed and logically presented, with adequate mathematical formalization.

•Parameter choices, computational aspects, and algorithmic flow are well explained, ensuring reproducibility.

Results and Discussion

•The experimental evaluation is comprehensive, demonstrating clear improvements over conventional PCA and related methods.

•Results are supported by appropriate benchmarking and statistical validation.

Conclusion

•The conclusion effectively summarizes the contributions and highlights the practical implications of the work.

•Future directions are appropriately suggested, adding value to the study.

Reviewer #3: This paper proposes a new clustering algorithm, H-NGPCA, for data streams. It combines three ideas: hierarchical structure, centroid-based clustering algorithm (NG), and online principal component analysis within each cluster/unit. The proposed algorithm is parameter-free in clustering, meaning it can adaptively decide the number of clusters and the number of components in each cluster. Experimental results demonstrate its advantages over the state-of-the-art cluserting algorithm.

Overall, this paper is well presented. However, I have the some comments for improvement in the attached pdf file.

6. PLOS authors have the option to publish the peer review history of their article (what does this mean?). If published, this will include your full peer review and any attached files.

Reviewer #1: No

Reviewer #2: No

Reviewer #3: No

---

## [Author Response · Author response to Decision Letter 1]

7 Nov 2025

We thank the reviewers and editor for their careful reading of our manuscript, and for their constructive comments.

We attached a file to provide a detailed response to each point raised.

---

## [Decision Letter · Decision Letter 1]

2 Dec 2025

H-NGPCA: Hierarchical clustering of data streams with adaptive number of clusters and adaptive dimensionality

PONE-D-25-43831R1

Dear Dr. Migenda,

We’re pleased to inform you that your manuscript has been judged scientifically suitable for publication and will be formally accepted for publication once it meets all outstanding technical requirements.

Kind regards,

Muhammad Ahsan, Ph.D.

Academic Editor

PLOS ONE

Additional Editor Comments (optional):

Reviewers' comments:

Reviewer's Responses to Questions

**Comments to the Author**

1. If the authors have adequately addressed your comments raised in a previous round of review and you feel that this manuscript is now acceptable for publication, you may indicate that here to bypass the “Comments to the Author” section, enter your conflict of interest statement in the “Confidential to Editor” section, and submit your "Accept" recommendation.

Reviewer #1: All comments have been addressed

Reviewer #2: All comments have been addressed

Reviewer #3: All comments have been addressed

2. Is the manuscript technically sound, and do the data support the conclusions?

Reviewer #1: Yes

Reviewer #2: Yes

Reviewer #3: Yes

3. Has the statistical analysis been performed appropriately and rigorously?

Reviewer #1: N/A

Reviewer #2: Yes

Reviewer #3: Yes

4. Have the authors made all data underlying the findings in their manuscript fully available?

Reviewer #1: Yes

Reviewer #2: Yes

Reviewer #3: Yes

5. Is the manuscript presented in an intelligible fashion and written in standard English?

Reviewer #1: Yes

Reviewer #2: Yes

Reviewer #3: Yes

6. Review Comments to the Author

Reviewer #1: All comments has been address. So, I recommend this paper publish in plos one as current version.

Reviewer #2: (No Response)

Reviewer #3: Good to see that the authors have incorporated the suggestions from me and the other two reviewers. It looks good to be published for me.

7. PLOS authors have the option to publish the peer review history of their article (what does this mean?). If published, this will include your full peer review and any attached files.

Reviewer #1: No

Reviewer #2: No

Reviewer #3: No

---

## [Editor Report · Acceptance letter]

PONE-D-25-43831R1

PLOS One

Dear Dr. Migenda,

I'm pleased to inform you that your manuscript has been deemed suitable for publication in PLOS One. Congratulations! Your manuscript is now being handed over to our production team.

Kind regards,

on behalf of

Dr. Muhammad Ahsan

Academic Editor

PLOS One